

# Relative errors of derived multi-wavelengths intensive aerosol optical properties using CAPS_SSA, Nephelometer and TAP measurements

Patrick Weber[1], Andreas. Petzold[1], Oliver F. Bischof[1,2], Benedikt Fischer[1], Marcel Berg[1], Andrew Freedman[3], Timothy Onasch[3], Ulrich Bundke[1]

[1]Forschungszentrum Jülich GmbH, Institute of Energy and Climate Research 8 – Troposphere (IEK-8), Jülich, Germany
[2]TSI GmbH, Particle Instruments, Aachen, Germany
[3]Aerodyne Research Inc., Billerica, MA 01821, USA

*Correspondence to*: Ulrich Bundke (u.bundke@fz-juelich.de)

**Abstract.** Aerosol intensive optical properties like the Ångström exponents for aerosol light extinction, scattering and absorption, or the single-scattering albedo are indicators for aerosol size distributions, chemical composition and radiative behaviour and contain also source information. The observation of these parameters requires the measurement of aerosol optical properties at multiple wavelengths which usually implies the use of several instruments. Our study aims to quantify the uncertainties of the determination of multiple-wavelengths intensive properties by an optical closure approach, using different test aerosols. In our laboratory closure study, we measured the full set of aerosol optical properties for a range of light-absorbing aerosols with different properties, mixed externally with ammonium sulphate to generate aerosols of controlled single-scattering albedo. The investigated aerosol types were: fresh combustion soot emitted by an inverted flame soot generator (SOOT, fractal aggregates), Aquadag (AQ, spherical shape), Cabot industrial soot (BC, compact clusters), and an acrylic paint (Magic Black, MB). One focus was on the validity of the Differential Method (DM: absorption = extinction minus scattering) for the determination of Ångström exponents for different particle loads and mixtures of light-absorbing aerosol with ammonium sulphate, in comparison to data obtained from single instruments. The instruments used in this study were two CAPS PM$_{ssa}$ (Cavity Attenuated Phase Shift Single Scattering Albedo, $\lambda = 450, 630$ nm) for light extinction and scattering coefficients, one Integrating Nephelometer ($\lambda = 450, 550, 700$ nm) for light scattering coefficient and one Tricolour Absorption Photometer (TAP, $\lambda = 467, 528, 652$ nm) for filter-based light absorption coefficient measurement. Our key finding is that the coefficients of light absorption $\sigma_{ap}$, scattering $\sigma_{sp}$ and extinction $\sigma_{ep}$ from the Differential Method agree with data from single reference instruments, and the slopes of regression lines equal unity within the precision error. We found, however, that the precision error for the DM exceeds 100% for $\sigma_{ap}$ values lower than 10-20 Mm$^{-1}$ for atmospheric relevant single scattering albedo. This increasing uncertainty with decreasing $\sigma_{ap}$ yields an absorption Ångström exponent (AAE) that is too uncertain for measurements in the range of atmospheric aerosol loadings. We recommend using DM only for measuring AAE values for $\sigma_{ap} > 50$ Mm$^{-1}$. Ångström exponents for scattering and extinction are reliable for extinction coefficients from 20 up to 1000 Mm$^{-1}$ and stay within 10% deviation from reference instruments, regardless of the chosen method. Single-scattering albedo (SSA) values for 450 nm and 630 nm wavelengths agree with values from the reference method $\sigma_{sp}$ (NEPH)/$\sigma_{ep}$ (CAPS PM$_{SSA}$)





with less than 10% uncertainty for all instrument combinations and sampled aerosol types which fulfil the defined goals for measurement uncertainty of 10% proposed by Laj et al., 2020 for GCOS (Global Climate Observing System) applications.


## 1. Introduction

The precise determination of aerosol optical properties is crucial for the provision of reliable input data for chemistry transport models, climate models, and radiative forcing calculations (Myhre et al., 2013). This applies in particular to light-absorbing

particles like black carbon (Petzold et al., 2013), which are produced by incomplete combustion processes and absorb visible light very efficiently. Aerosol light absorbing properties are also relevant for source appointment studies and the determination of anthropogenic influences on the atmospheric aerosol (Sandradewi et al., 2008) . There are two common methods to generate aerosol light absorption data for long-term and short-term measurements, each with its own disadvantages. One method is a filter-based technique, which operates by deriving light absorbing values from the attenuation of light trough particle-loaded

filter (Rosen et al., 1978). A disadvantage of all filter-based methods is linked to effects like multiple scattering inside the filter matrix, shadowing of light-absorbing particles in highly loaded filters, and humidity effects (Moosmüller et al., 2009). Widely deployed filter-based light absorption measurement methods are the Particle Soot Absorption Photometer (PSAP: Bond et al., 1999) and its further development, the Tri-colour Absorption Photometer (TAP: (Ogren et al., 2017), the Aethalometer (Hansen et al., 1984), and the Multi-Angle Absorption Photometer (MAAP) (Petzold et al., 2005). Except for the MAAP, all filter-

based methods require complex correction algorithms (Collaud Coen et al., 2010; Virkkula, 2010). Another method for deriving light absorption coefficients is the differential method, based on the subtraction of light scattering from light extinction coefficients. This method is commonly conducted by comparing measurements from two separate instruments which results in large precision errors particularly for lower aerosol light absorption coefficients. In laboratory studies, however, the differential method is widely used as reference technique because the applied light scattering and extinction instruments are

well characterised (Bond et al., 1999; Schnaiter et al., 2005; Sheridan et al., 2005). A significant improvement of aerosol measurement capacities is achieved by the recently developed Cavity Attenuated Phase Shift particle monitor for single scattering albedo (CAPS PM$_{SSA}$) (Onasch et al., 2015b) which is able to measure light extinction and scattering simultaneously and is the focus of recent studies (Perim de Faria et al., 2021; Modini et al., 2021) .

Intensive aerosol parameters like the Single Scattering Albedo (SSA) or Ångström exponents are often not directly measured,

but calculated from multiple instrument datasets, which could lead to an increase in errors and uncertainties concerning this parameter. The importance of reliable intensive parameters is undisputable, especially, when the use of them is required for an experiment or sensitive climate related modelling. The Ångström exponents are widely used to adjust extensive parameters to a desired wavelength (Ångström, 1929); Foster et al. (2019) for instrument comparison and more importantly for aerosol characterisation (Russell et al., 2010) like the refraction index calculation of mineral dust (Petzold et al., 2009) or black carbon

(Kim et al., 2015), or for source identification of mineral dust (Formenti et al., 2011). The scattering Ångström exponent (SAE) is size-dependent and therefore, used as an indication of the size distribution of aerosols in the investigated medium. The SAE





value of 4 indicates either a gaseous medium or a medium with nanometer-sized particles, whereas a value of 0 indicates coarse particles (Kokhanovsky, 2008). The absorption Ångström exponents (AAE) depends on the chemical composition of the aerosol. A value of 1 indicates an aerosol which absorbs light strongly across the entire visible spectral range and is composed

of nanometer-sized spheres (Berry and Percival, 1986). This behaviour is characteristic for fresh soot or black carbon fractal agglomerates (Kirchstetter and Thatcher, 2012; Xu et al., 2015). AAE values higher than unity indicate the presence of brown carbon (Kim et al., 2015) or mineral dust (Formenti et al., 2011), both of which are characterised by a stronger absorption in the blue and ultraviolet compared to the red spectral range. The extinction Ångström exponent (EAE) is often used for aerosol classification by remote sensing methods such as Lidar and depends on size distribution and chemical composition (Kaskaoutis

et al., 2007; Veselovskii et al., 2016). Combining those exponents in a cluster plot is a reliable method for classify aerosol sources (Russell, 2010). The SSA of an aerosol is the key parameter for its direct and semi direct impact on the climate (Penner, 2001). It describes the ratio of scattering to total extinction of a medium. The value of 1 indicates that light extinction relies exclusively on light scattering. In contrast, low SSA values indicate an aerosol with a large fraction of light-absorbing components, which may cause heating of the atmosphere. The intensive parameters are only available through multiple-

instrument approaches at different wavelengths which calls for a detailed analysis of measurement uncertainties. Our study contributes to this topic with a detailed optical closure study in which we deploy standard and advanced instrumentation for measuring aerosol optical properties and sample mixtures of light absorbing and scattering aerosol to assess method uncertainties and precision errors.


## 2. Experimental Approach

### 2.1 Experimental Design

In this study, we combined the use of different instruments with various aerosol types. In order to minimize instrument and measurement errors, a couple of preparations were necessary. For example, we ensured that the aerosol production is operated

using constant volumetric air flow. Also Ammonium-sulphate concentrations used where not changed during the experiment. Otherwise, this would change the particle size and thus the size distribution, leading to a less-well defined aerosol. Every measurement was done under ambient conditions in the lab. This was monitored by the internal pressure and temperature sensors of the nephelometer.





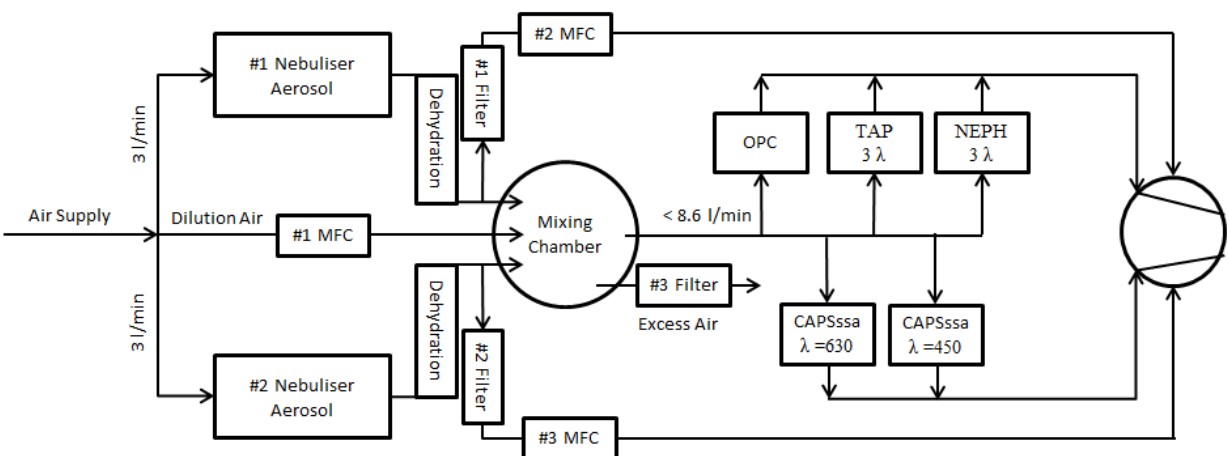

**Figure 1.** Experimental setup for the initial measurements.

The aerosol production was controlled by multiple Mass Flow Controllers (MFC, Bronkhorst High-Tech B.V., Ruurlo, Netherlands). A Labview based program controlled the complete measurement system and recorded centrally all data from the individual instruments. Downstream the production the aerosol was injected in a mixing chamber assuring homogenous mixing. The mixing chamber is attached to the aerosol supply line. Several instruments are connected to the central aerosol supply line where the individual instruments are connected to using an iso-axial orientated and isokinetic operated nozzle located in the centreline of the supply line. A Grimm optical particle size spectrometer (SKY-OPC, model 1.129, Grimm Aerosol GmbH & Co. KG, Ainring, Germany) was used to characterize and monitor the resulting size distribution. The particle scattering coefficient $\sigma_{sp}$ was measured with a integrating multi wavelength nephelometer (Model 3563, TSI Inc., Shoreview, MN, USA) (Bodhaine et al., 1991) and by a integrating sphere used in the CAPS PM$_{SSA}$ monitor (CAPS PM_SSA, Aerodyne Research Inc., Billerica, MA, USA; Onasch et al. (2015)). For the particle light absorption coefficient $\sigma_{ap}$ we used the small sized TAP (Brechtel Inc., Hayward, CA, USA) based on the well-known Particle Soot Absorption PSAP and the Continuous Light Absorption Photometer (CLAP) developed by NOAA (Ogren et al., 2017). The particle light extinction coefficient $\sigma_{ep}$ was directly measured with the phase shift channel of the CAPS PM$_{SSA}$ monitor.

All tubes and connections after the nebulizer were made of stainless steel or conductive silicone tubing to reduce particle loss by electrostatic forces. The humidity rarely exceeded 7%, which was an additional parameter measured by the nephelometer, so deliquescence effects were avoided. Because all instruments were connected to one central aerosol supply line. It was necessary to reduce the air flow towards the nephelometer from 20 l/min to 2.2 l/min due to the flow limits of the aerosol production. The flow-range of the other instruments, span from 0.6 l/min to 3 l/min. Due to the reduced air flow of the nephelometer also the time resolution of the nephelometer was reduced due to the longer flushing time of around 10 minutes.



### 2.2 Data analysis

*2.2.1 Corrections and calibrations*

The CAPS PM$_{SSA}$ instrument extinction channel was calibrated with polystyrene latex beads (PSL) particles as reference and Mie theory using BHMIE Python code derived from Bohren & Hoffman (1983). Additionally, the 450 nm wavelength CAPS PM$_{SSA}$ was calibrated with $CO_2$ for additionally validating the same factor and the calibration was applied to the nephelometer (Anderson and Ogren, 1998; Modini et al., 2021). The scattering channel of the CAPS PM$_{SSA}$ using the integrating sphere

method was internally adjusted to the extinction channel using ammonium sulphate as a light-scattering aerosol assuming a single scattering albedo of 1. A truncation error correction was not necessary regarding the size of the aerosols used (Onasch et al., 2015a) since highest amount of aerosols were smaller than 200 nm in diameter size. The CAPS PM$_{SSA}$ has a drifting shift of the base line as long the system is heating up, which apparently stabilized after 30 min of operation (Faria et al., 2019). The nephelometer (NEPH) correction for light absorbing aerosols was calculated according to (Massoli et al., 2009). Because

of the reduced air flow, the nephelometer needed at least 15 minutes to reach a stable plateau after changing aerosol production settings. After that, a new Filter Spot for the TAP was selected, to minimize transmission uncertainties increases by loaded filters.

**Table 1.** List of applied correction algorithms to optical instruments.

| Instrument | Manufacturer | Properties | λ (nm) | Reference |
|---|---|---|---|---|
| CAPS PM$_{SSA}$ | Aerodyne Research Inc. | $\sigma_{ep}$ ;$\sigma_{sp}$ | 450; 630 | Onasch et al. (2015) |
| NEPH | TSI Inc. | $\sigma_{sp}$ | 450; 550;700 | Anderson and Ogren (1998) ; Massoli et al (2009) |
| TAP | Brechtel Inc. | $\sigma_{ap}$ | 467; 530; 660 | Virkkula (2010) |


Data inversion for the Nephelometer was done by correction of truncation effects which alterned the data of additionally 5% maximally. These corrections were either made using the approach proposed by Anderson and Ogren (1998) for purely scattering aerosols or by the approach suggested by Massoli et al. (2009) for the aerosol mixtures calculated with a real refraction index of 1.6. CAPS PM$_{ex}$ signals were used without further correction, except for the adjustment factor determined

by $CO_2$ measurements and PSL to MIE calculations.

Corrections of the TAP data were made according to (Virkkula, 2010). A new filter spot was selected for each measurement.

*2.2.2 Aerosol Optical Probertites derived from primary measurements*





The extensive parameters for aerosol light interactions are extinction, scattering and absorption. When two of them are known,

the missing one can be calculated with the help of this equation:

$$\sigma_{ep} = \sigma_{sp} + \sigma_{ap}$$    Eq. (1)

where $\sigma_{ep}$ is the extinction coefficient, $\sigma_{sp}$ the light scattering coefficient and $\sigma_{ap}$ the coefficient for light absorption by particles. The unit of all these parameters is Mm$^{-1}$ ("inverse Mega meters"; 1 Mm$^{-1}$ = 10$^{-6}$ m$^{-1}$).

Solving equation 1 for $\sigma_{ap}$ it is possible to derive the absorption coefficient by combining CAPS_SSA and nephelometer

measurements for comparison. In the following this will be called Differential Method (DM).

To calculate the Single Scattering Albedo (SSA), the particle light scattering must be divided by the particle light extinction:

$$(\lambda) = \frac{\sigma_{sp}}{\sigma_{ep}}$$    Eq. (2)

The Ångström exponents AE are calculated from:

$$AE = -\frac{\log\left(\frac{\sigma_p(\lambda 1)}{\sigma_p(\lambda 2)}\right)}{\log(\lambda 1/\lambda 2)}$$    Eq. (3)

By solving Eq. 3 for $\sigma_p(\lambda 1)$ and assuming a valid Ångström exponent the resulting equation (3a) is used for wavelength adjustments

$$\sigma_{xp}(\lambda 1) = \sigma_{xp}(\lambda 2) \cdot \left(\frac{\lambda 1}{\lambda 2}\right)^{-AE}$$    Eq. (3a)

For the particle coefficient $\sigma_{xp}$ the corresponding $\sigma_{sp}$, $\sigma_{ep}$ or $\sigma_{ap}$ could be put into calculations (Eq. 3) to obtain the absorption Ångström exponent (AAE), extinction Ångström exponent (EAE) and scattering Ångström exponent (SAE) accordingly.

*2.2.3 Error propagation*

Error propagation for precision errors Δ are determined by Gaussian error propagation:

$$SSA(\lambda, \sigma_{sp}, \sigma_{ep}) = \frac{\sigma_{sp}}{\sigma_{ep}} \xrightarrow{yields} \Delta SSA(\lambda, \sigma_{sp}, \sigma_{ep}) = \sqrt{\left(\frac{1}{\sigma_{ep}} \cdot \Delta\sigma_{sp}\right)^2 + \left(\frac{\sigma_{sp}}{\sigma_{ep}^2} \Delta\sigma_{ep}\right)^2}$$    Eq. (4)

$$SSA(\lambda, \sigma_{sp}, \sigma_{ap}) = \frac{\sigma_{sp}}{\sigma_{ap}+\sigma_{sp}} \xrightarrow{yields} \Delta SSA(\lambda, \sigma_{sp}\sigma_{ap}) = \sqrt{\left(\frac{\sigma_{sp}}{(\sigma_{ap}+\sigma_{sp})^2} \cdot \Delta\sigma_{sp}\right)^2 + \left(\frac{\sigma_{ap}}{(\sigma_{ap}+\sigma_{sp})^2} \cdot \Delta\sigma_{ap}\right)^2}$$    Eq. (5)

$$AE = -\frac{\log\left(\frac{\sigma_{xp}(\lambda 1)}{\sigma_{xp}(\lambda 2)}\right)}{\log(\lambda 1/\lambda 2)} \xrightarrow{yields} \Delta AE = \sqrt{\left(\frac{-1}{\log(\lambda 1/\lambda 2) \cdot \sigma_p(\lambda 1)} \cdot \Delta\sigma_{xp}(\lambda 1)\right)^2 + \left(\frac{1}{\log(\lambda 1/\lambda 2) \cdot \sigma_{xp}(\lambda 2)} \cdot \Delta\sigma_p(\lambda 2)\right)^2}$$    Eq (6)

where $\sigma_{xp} = \{\sigma_{ep}, \sigma_{sp}, \sigma_{ap}\}$





### 2.3 Test Aerosol Generation

For every day of the experiments the solutions of Aquadag (AQ, Aqueous Deflocculated Acheson Graphite; Acheson
Industries, Inc., Port Huron, MI, USA), Cabot Black (BC) or the Acrylic Paint Magic Black (MB) were prepared by ultra-
sonication first, before nebulization in a Constant Output Atomizer (Model 3076, TSI Inc.). The resulting size for these
aerosols, as well as of the atomized ammonium sulphate, depended on the concentration put into solution and the air flow
rates. In order to vary the aerosol concentration with minimized sizes distribution chances, the mixture was controlled by a
MFC-determined active extractive flow after the dehydration tube. The inverted flame soot generator (Argonaut Scientific
Corporation, Edmonton, AB, Canada) was operated with a pre-determined propane to oxidation air ratio so that the flame
produced a stable and low organic carbon soot. It has previously been shown that at least 30 min were necessary to reach stable
aerosol concentrations (Bischof et al., 2019; Kazemimanesh et al., 2018)

**Table 2.** Overview of aerosol types used.

| Substance | Aerosol type | Acronym | Shape |
|---|---|---|---|
| Ammonium Sulphate | salt | AS | spheroidal shape |
| Aquadag | colloidal graphite | AQ | spherical shape |
| Cabot Black (R400R) | powder | BC | compact agglomerates |
| Flame Soot | combustion aerosol | Soot | fractal agglomerates |
| Magic Black (Acrylic paint) | dissolved pigments | MB | pigments |


With these sets of different aerosol types and shapes, the behaviour of instrument measurement is investigated. The results of
the intercomparison of Aquadag is expected to be best described by Mie theory, since its spherically shape and therefore
applied correction schemes to the instruments apply best, since calibration is done by ideal PSL spheres (polystyrene latex
beads), which were treated the same as all other aerosol solution samples and their size was approved by DMA and OPC.
Fractal agglomerates could have multiple internal scattering effects. Spherical shapes and several optical properties are
determined of the primary particle, this is expected to differ the most in intercomparison approaches (Barber and Wang, 1978;
Moosmüller et al., 2009).

**Table 3.** Overview of the used aerosol types and measured parameters.

| | AS | Magic Black | BC | AQ | Soot |
|---|---|---|---|---|---|
| | spheroid | acrylic paint | loose agglomerate | compact agglomerate | fractal structure |



| Median Diam. | 40 nm | 85 nm | 105 nm | 130 nm | 140 nm |
|---|---|---|---|---|---|
| Geometric standard deviation | 1.60 | 1.50 | 1.55 | 1.65 | 1.65 |
| SSA 630 (NEPH, CAPS) | 1.0 | $0.85 \pm 0.02$ | $0.26 \pm 0.03$ | $0.37 \pm 0.03$ | $0.20 \pm 0.02$ |
| SSA 450 (NEPH, CAPS) | 1.0 | $0.92 \pm 0.07$ | $0.32 \pm 0.04$ | $0.44 \pm 0.02$ | $0.26 \pm 0.08$ |
| SAE (630/450) (NEPH) | $3.22 \pm 0.09$ | $2.16 \pm 0.37$ | $1.71 \pm 0.13$ | $0.76 \pm 0.06$ | $0.99 \pm 0.08$ |
| AAE (630/450) (TAP) | - | $1.34 \pm 0.12$ | $1.16 \pm 0.03$ | $0.44 \pm 0.02$ | $1.08 \pm 0.02$ |
| EAE (630/450) (CAPS) | $3.21 \pm 0.08$ | $2.03 \pm 0.38$ | $1.43 \pm 0.65$ | $0.52 \pm 0.10$ | $1.10 \pm 0.10$ |

Table 3 shows the aerosol types used along with the measured size parameters and their calculated intensive parameters. The size distribution was measured beforehand with the combination of a Differential Mobility Analyzer (DMA 5.400, Grimm Aerosol Technik GmbH Co & KG Germany) and Condensation Particle Counter (CPC 5.411, Grimm Aerosol Technik) system in a sequential mode of operation. For internal calibration of the CAPS integrating sphere channel- measuring the light scattering coefficient- AS particles were used as purely scattering substance. By assuming a SSA of 1.0 the CAPS $PM_{SSA}$ Extinction channel is used as calibration reference.

The Ångström exponents for the pure substances are in typical ranges for these types of aerosols and size distributions reported in the literature. For example, the SAE decreases from a value of 3,22 for 40nm AS particles which is close to the SAE value of 4 for air molecules with increasing particle diameter. Thus, the SAE drops to 0,76 for 130 nm AQ particles. As expected by Eq. 3a the SSA increases with shorter wavelength (Bohren and Huffman, 1983). The AAE for fractal combustion soot is close to 1 as reported by e.g. (Török, 2018) for the mini-CAST soot generator.

The errors reported are either the instruments uncertainties or are calculated from error propagation. The light extinction channel of the CAPS instrument has an uncertainty of 5% and precision of 2% and a scattering uncertainty of 8% and 2% precision respectively (Onasch et al., 2015). The TAP has an uncertainty of around 8%, with a precision of 4% ((Müller et al., 2014; Ogren et al., 2017), while the nephelometer has an uncertainty of less than 10% and a precision of about 3% ((Anderson and Ogren, 1998), (Massoli et al., 2009).



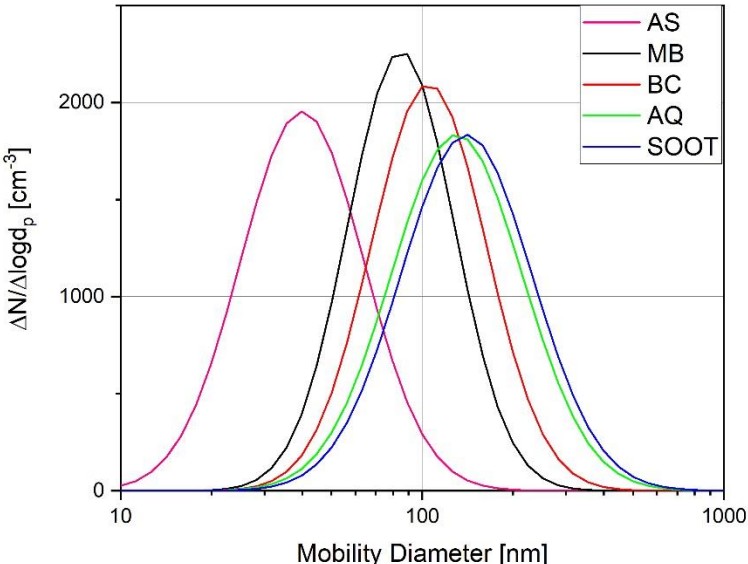


**Figure 2.** Measured Size Distributions by DMA and CPC for the aerosol types used, normalised to an assumed total concentration.

In order to give a brief overview of the test aerosol size distributions reported by the DMA and CPC system as a function of the electric mobility size diameter in nm, Figure 2 provides the size distributions of the different aerosol types normalized to

1000 particle counts (N) per cubic centimetre.

### 3. Measurements

In a first step, the extensive parameters must be validated for all instrument combinations to ensure the reliability of the

intensive parameters derived from them. Aquadag is well-known for its physical properties and it is easy to handle by nebulising. We have selected AQ as it is commonly used as a reference material for instrument comparisons (Foster et al., 2019) for all the viewgraphs. The results for the other aerosol types are added in the associated tables 6-9. Respective data points are given as averages of at least 100 seconds of stable aerosol production.

### 3.1 Extensive Parameters

The two CAPS_SSA monitors used measure the extinction coefficient of particles directly with a small precision error of around 2% (Modini et al., 2021) for 450nm and 630nm wavelength. In Figure 3 we show scatter plots of these direct



measurements (X-axis) in comparison to the combined measurements of the (absorption coefficient) using TAP and the

scattering coefficient using the nephelometer using Equation (Eq. 1) in the form: $\sigma_{ep}(NEPH, TAP) = \sigma_{ap}(TAP) + \sigma_{sp}(NEPH)$ (y-axis) for 450nm (right panel) and 630 nm wavelength (left panel)

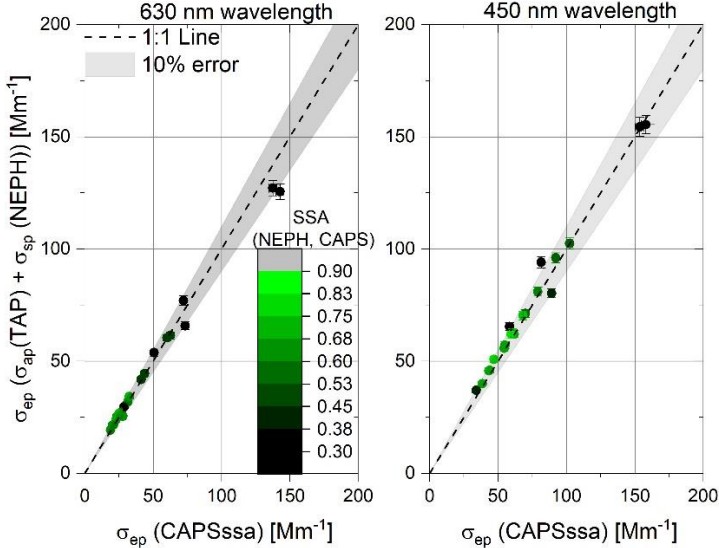

**Figure 3.** Scatter plot of the Extinction coefficients for different Aquadag- AS mixtures for 630 nm and 450 nm wavelengths measured by the combined TAP and Nephelometer data of absorption and scattering coefficients versus the CAPS_SSA

monitor direct extinction coefficient measurements. The colour code represents the SSA of the analysed mixed aerosol of the respective data point at 630 nm wavelength. In addition, an error band of ±10% was added to the 1:1 line.

Here mixtures of nebulized Aquadag particles and ammonium sulphate particles are used as a proxy for the mixing ratio the SSA is shown as colour code. The extinction coefficients align the 1:1 line within 10% in a broad range of the extinction

coefficient for 450 and 630 nm wavelength as well as for SSA of the mixtures ranging from 0.3 close to 1. This shows that the instruments are not sensitive to the SSA of the particle type used for both wavelengths of interest.

As the next extensive parameter, the scattering coefficient at 450 and 630 nm wavelengths are compared using scatterplots for different techniques in Figure 4. Here we use the Nephelometer and the integrating sphere channel of the CAPS_SSA

instrument capable of measuring the scattering coefficient directly. In addition we calculated the scattering coefficients using a Differential Method (DM) solving Eq.(1) for the scattering coefficient by subtracting the absorption coefficient measured by the TAP from the extinction coefficient measured by CAPS_SSA, The nephelometer is used as reference because it has well proven correction functions for light absorption particles, as described in Section 2.2.1.



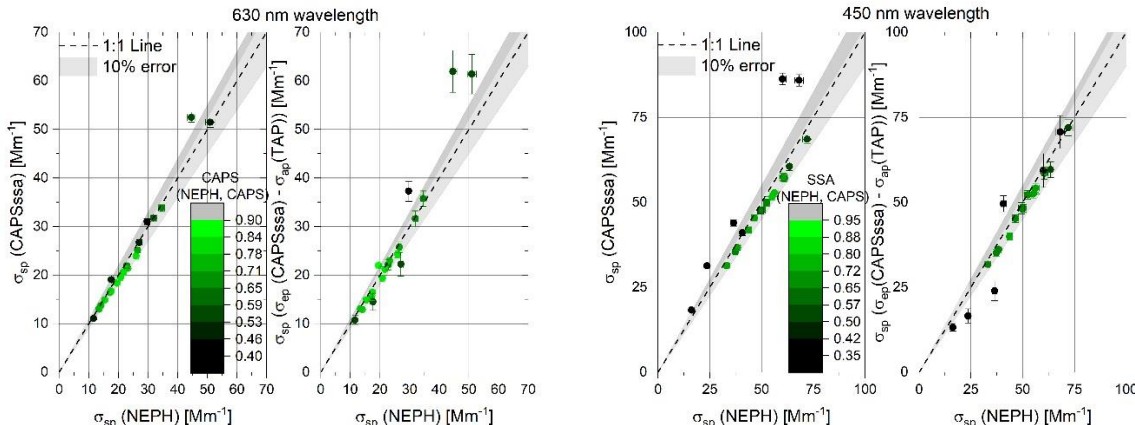

**Figure 4.** Comparison of Light scattering coefficients of mixtures of Aquadag with AS for 450nm and 630 nm wavelengths for Differential method (DM), CAPS_SSA (integrating sphere) techniques versus nephelometer measurements for 450 and 630nm wavelength using scatter plots. The colour code represents the SSA value of the measured aerosol mixture. An error band of ±10% was applied to the 1:1 line. Error bars shown represent one sigma of instrument precision.

The scattering coefficients agrees between the reference instrument in comparison to the internal scattering signal measured with the CAPS PM$_{SSA}$ and $\sigma_{sp}$ obtained by subtraction of $\sigma_{ap}$(TAP) from $\sigma_{ep}$(CAPS) shown in Figure 4 within 10% margin. There is neither a trend visible of the mixture ratio with ammonium sulphate, which the SSA is the indicator for, nor a strong shift for high or low volumetric cross-section values. This is true for both examined wavelengths of 630 nm and 450 nm. Overall, it is visible, that some data points scatter more roughly on the 1:1 line, which is true for mostly pure Aquadag aerosol, where the TAP contributes the biggest uncertainties due to higher values and deviates from high agreement, which was approved by the Reno Study (Sheridan, 2005) and although visible in our measurements. When the 1σ precision errors are tripled, it is still undistinguishable from the 1:1 line.

As a last extensive parameter, we focused on the particle light absorption coefficient. This is the most complicated to measure, as for filter-based methods a bunch of correction schemes must be applied. Using a differential method e.g. ($\sigma_{ap}$(CAPS,NEPH)= $\sigma_{ep}$(CAPS) – $\sigma_{sp}$(NEPH) following Eq.. 1) is used, we have to deal with large relative errors. Because of the availability of reference and calibration substances for filter based methods, $\sigma_{ap}$(CAPS,NEPH) is given as the reference for the comparison to the $\sigma_{ap}$(TAP) values and the internal $\sigma_{ap}$(CAPS,CAPS).





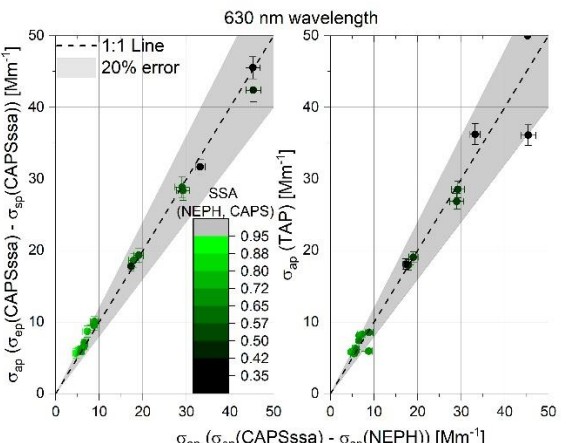
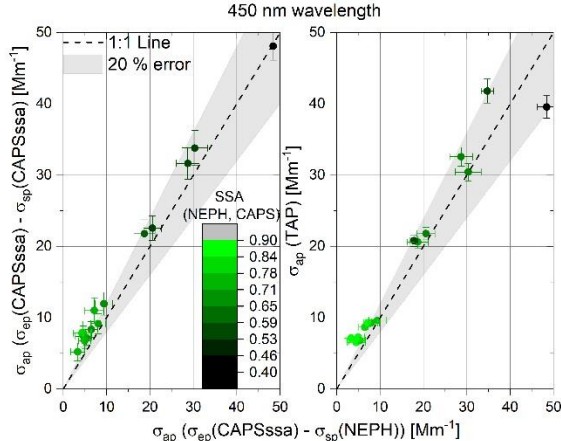

**Figure 5.** Scatter plots of absorption coefficients of Aquadag for 450 nm and 630 nm wavelengths for different instrument combinations. The colour code represents the SSA value of the respective data point. An error band of ±20% was applied to the 1:1 line. Individual error bars represent variances during on type of mixture produced.

In Figure 5, the light absorption values for wavelengths of 450 nm and 630 nm are depicted. To compare instruments, the overall uncertainty is often estimated to be 30% (Bond et al., 1999)In this work we stay within a 20% deviation for this parameter. Most data points correlated for both the $\sigma_{ap}$ (CAPS,CAPS) and $\sigma_{ap}$(TAP) reference, without any mixing ratio dependence. When the $\sigma_{ap}$(CAPS,CAPS)is compared to $\sigma_{ap}$(CAPS,NEPH), the values agree within the uncertainty errors.

**Table 4.** Linear regression analysis of attenuation coefficients for of Aquadag and ammonium sulphate mixtures given as slopes (m), Pearson R and y-axis intersection (b).

| | $\sigma_{sp\,(CAPS)}$ vs. $\sigma_{sp\,(NEPH)}$ | $\sigma_{sp(CAPS,TAP)}$ vs $\sigma_{sp(NEPH)}$ | $\sigma_{ep(NEPH,TAP)}$ vs. $\sigma_{ep(CAPS)}$ | $\sigma_{ap(TAP)}$ vs. $\sigma_{ap(CAPS,NEPH)}$ |
|---|---|---|---|---|
| | | | 630 nm | |
| m | $1.07 \pm 0.03$ | $1.08 \pm 0.05$ | $0.99 \pm 0.03$ | $0.92 \pm 0.07$ |
| R | 0.99 | 0.97 | 0.99 | 0.95 |
| b [Mm$^{-1}$] | $-1.84 \pm 0.57$ | $-2.15 \pm 1.12$ | $0.91 \pm 0.93$ | $0.78 \pm 0.68$ |
| | | | 450 nm | |
| m | $0.99 \pm 0.05$ | $1.06 \pm 0.03$ | $0.98 \pm 0.03$ | $1.04 \pm 0.08$ |
| R | 0.97 | 0.99 | 0.99 | 0.96 |





| b [Mm⁻¹] | 1.14 ± 2.27 | -4.60 ± 1.51 | 3.37 ± 1.71 | 2.13 ± 0.64 |

The high Pearson correlation (r > 0.95) coefficients in Table 4 indicate that the correlation is highly linear and reveals a stable behaviour of the instrument measurements characteristics. The slopes are all close to unity within the expected errors ranges.

Thus, the extensive parameters can be trusted for instrument comparison especially for the light scattering and light extinction information. The slopes reported for light absorption coefficients are with 0,92 ±0.07 and 1.04 ±0.08 below the expected error from literature. Higher values influence linear regression slopes, for which the filter methods are drifting to lower values respective to intercomparison instruments (Sheridan, 2005). We provide further regression analysis for all other aerosol types individually in Tables 7-9. An excellent agreement (r=0.99) is shown for $\sigma_{sp}$ measurements of the nephelometer and the CAPS

PM$_{SSA}$ scattering channel. Thus, the CAPS PM$_{SSA}$ gives reliable scattering coefficient measurements for aerosol mixtures and could be considered as a substitute for the nephelometer and delivers reliable SSA measured simultaneously in one volume of the same instrument.

Instead of using regression analysis, where outliers and/or high values are dominating the slope of the regression- a more robust statistical analysis of the ensemble averaged instrumental ratios ($\sigma_{ap}$ (instrument #1) / $\sigma_{ap}$ (instrument #2) will be shown

in the following section. For the table 5 – table 6 the ratios are calculated using averaged 1Hz measurement data. The average intervals are adapted for constant conditions, waiting 15 minutes until the production and nephelometer were settled/relaxed lasting for about 5 minutes until the next mixture was setup in the sample line restarting the procedure.

**Table 5.** Ensemble average as a ratio of $\sigma_{ap}$ (TAP) / $\sigma_{ap}$ (CAPS, NEPH) at 630 nm wavelength.

| 630 nm wavelength | BC | AQ | SOOT | MB |
|---|---|---|---|---|
| $\sigma_{ap}$ (TAP) / $\sigma_{ap}$ (CAPS,NEPH) with variance | 1.22 ± 2.57 (N=36) | 0.97 ± 0.22 (N=28) | 1.10 ± 1.22 (N=25) | 0.88 ± 0.17 (N=8) |
| $\sigma_{ap}$ (TAP) / $\sigma_{ap}$ (CAPS,NEPH) for samples with $\sigma_{ap}$ >10 Mm⁻¹ variance | 1.08 ± 0.19 (N=24) | 0.94 ± 0.10 (N=11) | 0.86 ± 0.13 (N=6) | - |


Table 5 demonstrate that the light absorption values agree for the different methods in general. With an ensemble average for the ratio $\sigma_{ap}$ (TAP) / $\sigma_{ap}$ (CAPS,NEPH) close to 1, a good agreement is achieved and over 60% of all datapoints for Aquadag fits within a range of $\sigma_{ap}$ (TAP) / $\sigma_{ap}$ (CAPS,NEPH) = {0.8 – 1.2} . Regarding fractal soot particles this ratio deviates most form 1, while using Cabot Black over 50% of all data to fit in the range of $\sigma_{ap}$ (TAP) / $\sigma_{ap}$ (CAPS,NEPH) = {0.8 – 1.2}.

Filtering these instrument ratios for $\sigma_{ap}$ < 10 Mm⁻¹ the relative frequency distribution shows almost no modal value. Filtering the data for $\sigma_{ap}$ > 10 Mm⁻¹ about 80% of these data are within the range of 0.8-1.2 .



**Table 6.** Ensemble averages as a ratio of $\sigma_{ap}$ (TAP) / $\sigma_{ap}$ (CAPS, NEPH) at 450 nm wavelength.

| 450 nm wavelength | BC | AQ | SOOT | MB |
|---|---|---|---|---|
| $\sigma_{ap}$ (TAP) / $\sigma_{ap}$ (CAPS,NEPH) with variance | 1.03 ± 1.72 (N=36) | 1.06 ± 0.38 (N=28) | 0.89 ± 1.05 (N=25) | 1.28 ± 2.91 (N=8) |
| $\sigma_{ap}$ (TAP) / $\sigma_{ap}$ (CAPS,NEPH) for samples with $\sigma_{ap}$ >10 Mm$^{-1}$ variance | 1.08 ± 0.33 (N=24) | 1.01 ± 0.13 (N=11) | 0.84 ± 0.27 (N=6) | - |

Redoing this analysis for 450 nm wavelength the light extinction and scattering of smaller particles increases compared to the values at 630nm wavelength. As a result, this increase also the errors associated with the differential method. As demonstrated in Table 6, only the ratio $\sigma_{ap}$ (TAP) / $\sigma_{ap}$ (CAPS, NEPH) for spherical particles deviate less from unity, with over 50% of the data being within the range of 0.8-1.2. Still all ensemble averages are close to 1 but with an associated error of up to ±1.7 these values are not significant, which means, that the ratios scatter widely with no clear modal value.

Again filtering only for $\sigma_{ap}$ >10 Mm-1 the methods agree well with significant ratios $\sigma_{ap}$ (TAP) / $\sigma_{ap}$ (CAPS, NEPH) =1.08 ± 0.33 for BC. The best instrumental ratio with 1.01 ± 0.13 is shown for AQ in Table 6 at 450 nm wavelength.



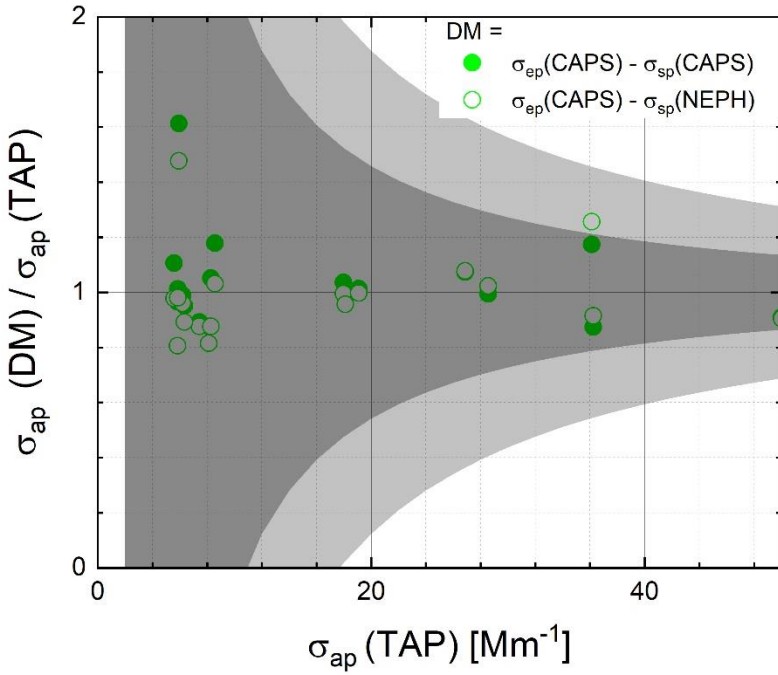


**Figure 6.** For Both differential methods calculation the absorption coefficient $\sigma_{ap}$ (DM)= {$\sigma_{ap}$ (CAPS,CAPS) see filled symbols, $\sigma_{ap}$ (CAPS,NEPH) see open symbols} the rations of $\sigma_{ap}$ (DM) / $\sigma_{ap}$ (TAP) Aquadag aerosol type mixtures as function of $\sigma_{ap}$(TAP) are shown. The dark grey error band represent the calculated relative errors using Gaussian error propagation assuming $\sigma_{ep}$=50 Mm$^{-1}$; the light grey error band represents calculated relative errors assuming constant $\sigma_{ep}$ = 200 Mm$^{-1}$.


In order to demonstrate the dependency on the magnitude of $\sigma_{ap}$ the instrumental ratios of $\sigma_{ap}$ (TAP) / $\sigma_{ap}$ (CAPS, NEPH) and $\sigma_{ap}$ (TAP) / $\sigma_{ap}$ (CAPS, CAPS) are shown as function of $\sigma_{ap}$ (TAP) in Figure 6. For $\sigma_{ap}$ values lower than 20 Mm$^{-1}$, errors even over 100% for light absorption coefficient are derived for the DM methods assuming for $\sigma_{ep}$ = 200 Mm$^{-1}$. The experimental

data stays within this calculated relative uncertainty. For $\sigma_{ap}$ values over 50 Mm$^{-1}$, the instrumental ratio deviates from 1 less than 10-20%. Both differential methods show an excellent agreement as already demonstrated in Figure 5 thus, open and filled marks representing the two different methods are always in close proximity.


**Table 7.** Linear regression analysis of attenuation coefficients using Cabot Black and ammonium sulphate mixtures are shown. Presenting: the slope (m), Pearson (r) and y-axis intersection (b) for different instruments combinations.





| BC | $\sigma_{sp}(CAPS)$ vs $\sigma_{sp}(NEPH)$ | $\sigma_{sp(CAPS,TAP)}$ vs. $\sigma_{sp(NEPH)}$ | $\sigma_{ep(TAP,NEPH)}$ vs $\sigma_{ep(CAPS)}$ | $\sigma_{ap(TAP)}$ vs. $\sigma_{ap(CAPS,NEPH)}$ |
|---|---|---|---|---|
| | | 630 nm | | |
| m | $1.02 \pm 0.03$ | $0.99 \pm 0.05$ | $0.94 \pm 0.02$ | $0.90 \pm 0.02$ |
| r | 0.98 | 0.96 | 0.99 | 0.99 |
| b [Mm$^{-1}$] | $-0.69\pm0.7$ | $-2.13 \pm 1.01$ | $3.59 \pm 0.60$ | $2.57 \pm 0.11$ |
| | | 450 nm | | |
| m | $0.99 \pm 0.02$ | $1.06 \pm 0.06$ | $0.94 \pm 0.03$ | $0.86 \pm 0.05$ |
| r | 0.99 | 0.95 | 0.98 | 0.97 |
| b [Mm$^{-1}$] | $5.36 \pm 1.45$ | $-0.59 \pm 3.86$ | $0.97 \pm 3.17$ | $2.98 \pm 0.48$ |

**Table 8.** Linear regression analysis attenuation coefficients slopes for fresh combustion soot and ammonium sulphate mixtures given as slopes (m), Pearson (r) and y-axis intersection (b).

| SOOT | $\sigma_{sp}(CAPS)$ vs $\sigma_{sp}(NEPH)$ | $\sigma_{sp(CAPS,TAP)}$ vs. $\sigma_{sp(NEPH)}$ | $\sigma_{ep(TAP,NEPH)}$ vs $\sigma_{ep(CAPS)}$ | $\sigma_{ap(TAP)}$ vs. $\sigma_{ap(CAPS,NEPH)}$ |
|---|---|---|---|---|
| | | 630 nm | | |
| m | $1.06 \pm 0.04$ | $0.9 \pm 0.20$ | $0.99 \pm 0.08$ | $0.76 \pm 0.11$ |
| r | 0.99 | 0.74 | 0.97 | 0.92 |
| b [Mm$^{-1}$] | $0.05 \pm 0.56$ | $1.57 \pm 3.21$ | $1.80 \pm 1.72$ | $3.93 \pm 1.68$ |
| | | 450 nm | | |
| m | $0.81 \pm 0.03$ | $0.77 \pm 0.07$ | $0.92 \pm 0.04$ | $0.70 \pm 0.10$ |
| r | 0.99 | 0.97 | 0.98 | 0.91 |
| b [Mm$^{-1}$] | $1.73 \pm 0.45$ | $2.64 \pm 0.91$ | $3.26 \pm 2.24$ | $1.75 \pm 0.82$ |

**Table 9.** Linear regression analysis of volumetric cross sections slopes of the acrylic paint (Magic Black (MB)) and ammonium sulphate mixtures given as slopes, Pearson R and y-axis intersection.

| MB | $\sigma_{sp}(CAPS)$ vs $\sigma_{sp}(NEPH)$ | $\sigma_{sp(CAPS,TAP)}$ vs. $\sigma_{sp(NEPH)}$ | $\sigma_{ep(TAP,NEPH)}$ vs $\sigma_{ep(CAPS)}$ | $\sigma_{ap(TAP)}$ vs. $\sigma_{ap(CAPS,NEPH)}$ |
|---|---|---|---|---|





| | 630 nm | | | |
|---|---|---|---|---|
| m | 0.96 ± 0.03 | 1.05 ± 0.03 | 0.96 ± 0.03 | 0.57 ± 0.10 |
| r | 0.99 | 0.99 | 0.99 | 0.94 |
| b [Mm$^{-1}$] | 0.42 ± 0.79 | -0.95 ± 0.53 | 0.99 ± 0.51 | 1.06 ± 0.38 |
| | 450 nm | | | |
| m | 1.02 ± 0.02 | 1.00 ± 0.16 | 0.89 ± 0.11 | 0.21 ± 0.14 |
| r | 0.99 | 0.95 | 0.97 | 0.58 |
| b [Mm$^{-1}$] | -1.85 ± 0.78 | -0.82 ± 6.04 | 4.58 ± 4.88 | 3.43 ± 0.91 |

The linear regression analysis reporting slopes, Pearson coefficients and offsets for attenuation coefficients for the different light absorbing aerosol types are presented in Table 7 (BC), Table 8 (soot), and Table 9 (MB). In general, for 630 nm
wavelength high Pearson rates (r>0,96) with negligible offsets (b<1 Mm$^{-1}$) and slopes ranging from 0.90 to 1.05 demonstrates a good agreement for scattering and extinction coefficient measurements. Especially for MB and Soot the Differential method tends to underestimate the value compared to TAP measurements by 20- 40 % whereas for BC the difference is only 10% The reason could be that soot is a fractal agglomerate and in-situ methods as well as filter-based methods give different results as a function of the primary particle size (Sorensen et al., 2010p) as well of the previous mentioned changes of the slope at higher
$\sigma_{ap}$ (TAP) values.

For 450 nm wavelength similar slopes, Person and offset values could be reported for these aerosol types mixtures with ammonium sulphate. Fresh soot particle mixtures values decreasing with the lower wavelength to a slope of 0.77 for light scattering intercomparison and $\sigma_{ap}$ of 0.7. These is as well an effect of the primary particles size of agglomeration, since those relationship chances with the wavelength.

For Magic Black the light absorption measurements using the DM method for 450nm shows the highest difference compared to the TAP measurement with a regression slope of 0.21 ± 0.14. The reason could be different absorption behaviour for in-situ to filter measurements and no clear indications of the particle shape could be made.

## 2.2 Intensive Parameters

2.2.1 Single scattering Albedo (SSA)

The Single Scattering Albedo (SSA) as an important climate parameter is used for instrument validation in this section. To obtain this parameter, different methods are shown in Table 10. Each method excludes at least one instrument from the calculation, thus, instrument intercomparison is possible.





The SSA for different combinations are derived using Eq. (2) as follows. In the following the instrument used are denoted in
parentheses.

$$SSA(NEPH, TAP) = \frac{\sigma_{sp}(\text{NEPH})}{\sigma_{ap}(\text{TAP}) + \sigma_{sp}(\text{NEPH})} \qquad\qquad \text{Eq. (7)}$$

$$SSA(CAPS, TAP) = \frac{\sigma_{ep}(\text{CAPS}) - \sigma_{ap}(\text{TAP})}{\sigma_{ep}(\text{CAPS})} \qquad\qquad \text{Eq. (8)}$$


$$SSA(CAPS, CAPS) = \frac{\sigma_{sp}(\text{CAPS})}{\sigma_{ep}(\text{CAPS})} \qquad\qquad \text{Eq. (9)}$$

As reference we use the often-used combination:

$$SSA(NEPH, CAPS) = \frac{\sigma_{sp}(\text{NEPH})}{\sigma_{ep}(\text{CAPS})} \qquad\qquad \text{Eq. (10)}$$



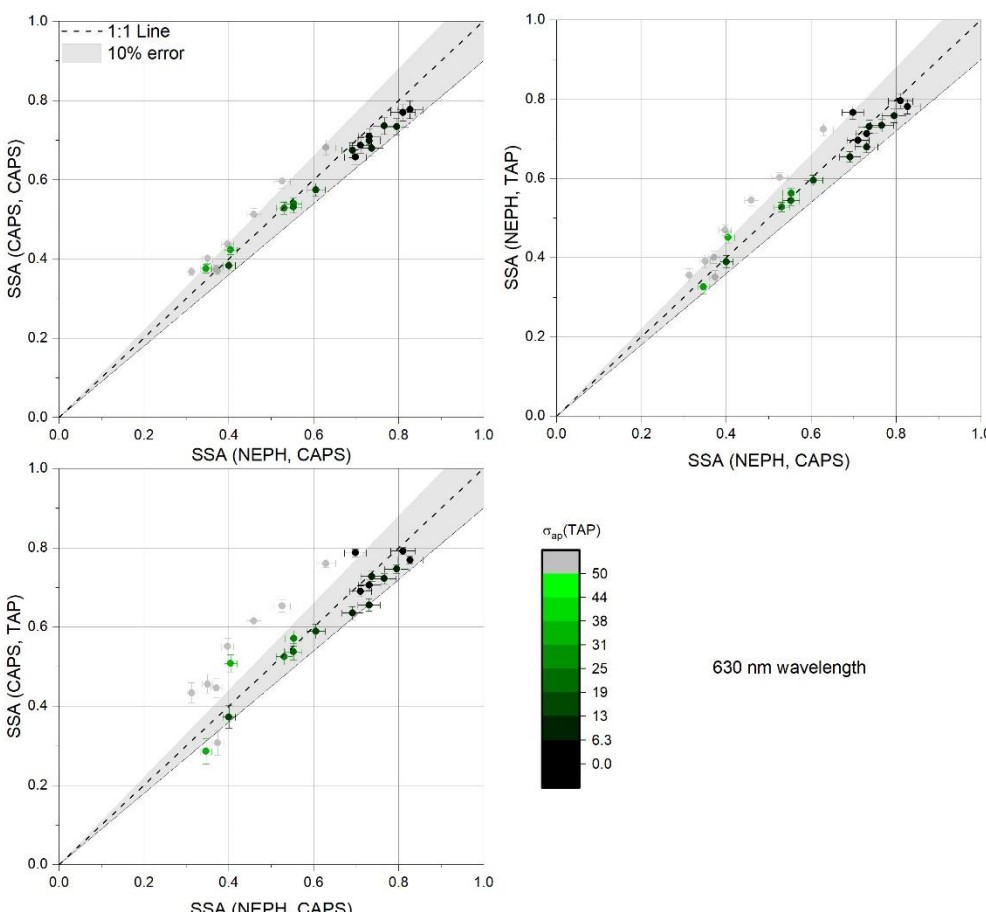

**Figure 7.** This figure shows scatter plots of SSA Instrument-to-instrument measurement ratios for 630 nm wavelength reported
for AQ/AS mixtures (y-axis) versus SSA(NEPH,CAPS)
as reference on the x-axis. (see Equations 7-11. The colour code indicates $\sigma_{ap}$(TAP) values shown in Mm$^{-1}$.

Figure 7 shows the SSA parameters obtained by the three combination of instruments for 630 nm wavelength. The correlation
shows reasonable results within a +-10% error band. Spotting for dark colours in the colourcode - Low $\sigma_{ap}$ values are only seen
for SSA >0.6 as expected reflecting that there are just fewer particles of Aquadag in the aerosol mixture. General, the load of
absorbing particles seems not to influence the accuracy of the method, except for high absorption coefficients over 50 Mm$^{-1}$.
Here the TAP shows a nonlinear response which is visible in Figure 7 as offset of 0.1 higher than the SSA reference calculated
using Eq. (10).



Like in the previous section, the ensemble average (average of instrument-to-instrument measurement ratios) was calculated
to show a robust measure for the overall agreement of this parameter. In Table 10 the SSA values for all aerosol types are
summarized. The nephelometer and CAPS extinction was used again as reference. The highest deviation is visible with
combustion soot for TAP related data. The deviations of the reported mean from 1 are less than the relative uncertainties which
range around 0.09.


**Table 10.** Ensemble average of instrument-to-instrument measurement ratios for different instrument combination to obtain
the SSA at 630 nm wavelength using $SSA(NEPH, CAPS)$ as reference

| Instrument combination used for SSA calculation | BC | AQ | SOOT | MB |
|---|---|---|---|---|
| $SSA\ (CAPS,\ CAPS)$ | $1.00 \pm 0.08$ | $1.01 \pm 0.07$ | $1.07 \pm 0.07$ | $1.00 \pm 0.04$ |
| $SSA(NEPH, TAP)$ | $0.96 \pm 0.08$ | $1.02 \pm 0.08$ | $1.04 \pm 0.29$ | $1.00 \pm 0.03$ |
| $SSA(CAPS, TAP)$ | $0.98 \pm 0.16$ | $1.05 \pm 0.16$ | $1.07 \pm 0.51$ | $1.00 \pm 0.03$ |





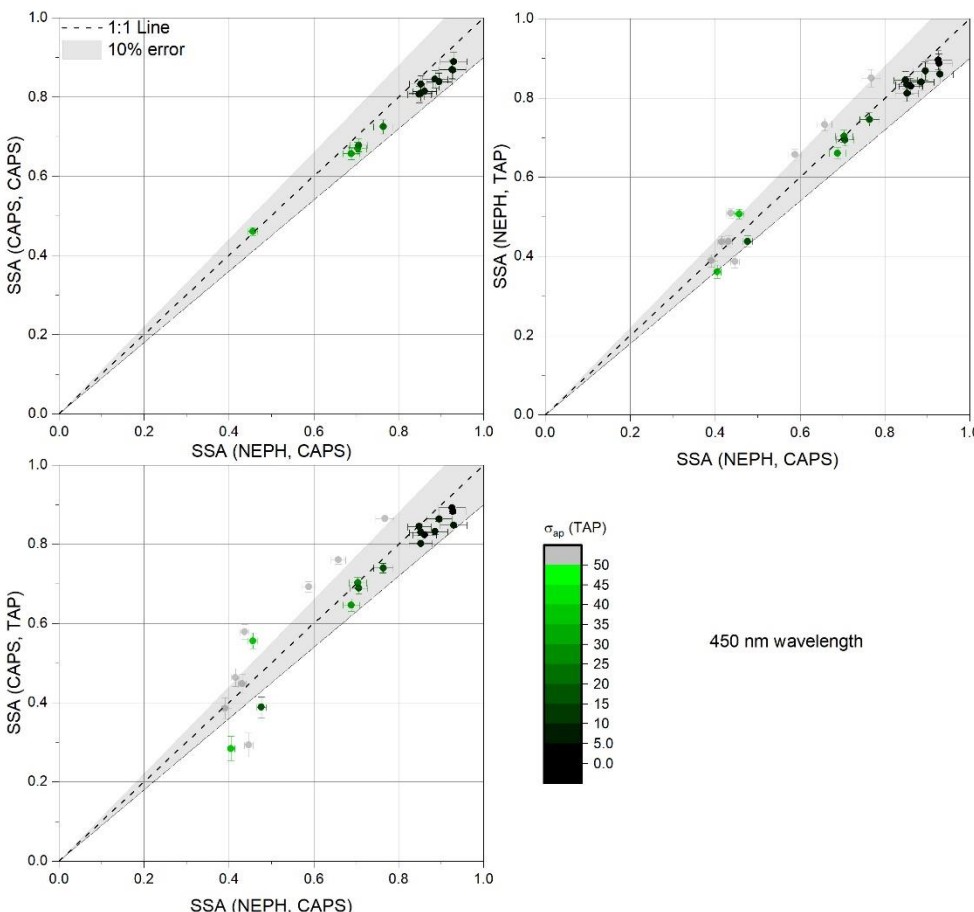

**Figure 8.** This figure shows scatter plots of Instrument-to-instrument ratios for the SSA for 450 nm wavelength using AQ/AS mixtures for different instrument combinations as function of the reference SSA (NEPH, CAPS). The colour code indicates $\sigma_{ap}$(TAP) values.

Figure 8 present scatter plots of Instrument-to-instrument ratios for the SSA values for 450 nm wavelength using AQ/AS mixtures for all instrument combinations. Observed patters are comparable to the results of Figure 7 for 630 nm wavelength. For absorption coefficients up to 50 Mm$^{-1}$ all methods agree within 10%. Above 50 Mm$^{-1}$ again the non-linear response of TAP is visible again showing an offset of 0.1 for the instrument-to-instrument ratio.

**Table 11.** Ensemble averages of instrument-to-instrument measurement ratios for different instrument combination to obtain the SSA at 450 nm wavelength using $SSA(NEPH, CAPS)$ as reference (see Eq. 7-10).



| Instrument combination used for SSA calculation | BC | AQ | SOOT | MB |
|---|---|---|---|---|
| SSA (CAPS, CAPS) | 1.17 ± 0.21 | 1.04 ± 0.13 | 1.11 ± 0.13 | 0.98 ± 0.02 |
| SSA (NEPH, TAP) | 1.07 ± 0.08 | 1.02 ± 0.08 | 0.96 ± 0.19 | 1.04 ± 0.13 |
| SSA (CAPS, TAP) | 1.11 ± 0.13 | 1.03 ± 0.14 | 0.64 ± 0.38 | 1.05 ± 0.14 |

The pattern, that more fractal aerosols differ from the ensemble average as the wavelength decreases is visible here too. The fresh combustion soot aerosol shows with 0.64 ± 0.38 highest deviation from 1 for SSA (CAPS, TAP). But overall, all instrument-to-instrument ratios are close to unity within the observed variance.


### 2.2.2 Ångström exponents: EAE

In this section we will now focus on the next important and climate model-relevant aerosol parameter. The Ångström exponents are calculated from extensive parameters of different wavelengths. Even a small precision error results in a high deviation,
considering error propagation. To demonstrate the overall variability, all aerosol types measured are shown simultaneously in all plot in this section.

The following equations based on Eq.(3) are used to derive the Angström exponents for extinction, scattering and absorption using different Instrument combinations:


$$AE(\text{Instrument1}, \text{Instrument 2}) = -\frac{\log\left(\frac{\sigma_{xp\lambda1}(\text{Instrument1}, \text{Instrument 2})}{\sigma_{xp\lambda2}(\text{Instrument1}, \text{Instrument 2})}\right)}{\log(\lambda1/\lambda2)} \qquad \text{Eq. (11)}$$

$$EAE(\text{CAPS}) = -\frac{\log\left(\frac{\sigma_{ep\lambda1}(\text{CAPS})}{\sigma_{ep\lambda2}(\text{CAPS})}\right)}{\log(450/630)} \qquad \text{Eq. (12)}$$

$$EAE(\text{NEPH}, \text{TAP}) = -\frac{\log\left(\frac{\sigma_{ep\lambda1}(\sigma_{ap}(\text{TAP})+\sigma_{sp}(\text{NEPH}))}{\sigma_{ep\lambda2}(\sigma_{ap}(\text{TAP})+\sigma_{sp}(\text{NEPH}))}\right)}{\log(450/630)} \qquad \text{Eq. (13)}$$

$$SAE(\text{NEPH}) = -\frac{\log\left(\frac{\sigma_{sp\lambda1}(\text{NEPH})}{\sigma_{sp\lambda2}(\text{NEPH})}\right)}{\log(450/700)} \qquad \text{Eq. (14)}$$

$$SAE(\text{CAPS}, \text{TAP}) = -\frac{\log\left(\frac{\sigma_{sp\lambda1}(\sigma_{ep}(\text{CAPS})-\sigma_{ap}(\text{TAP}))}{\sigma_{sp\lambda2}(\sigma_{ep}(\text{CAPS})-\sigma_{ap}(\text{TAP}))}\right)}{\log(450/630)} \qquad \text{Eq. (15)}$$

$$AAE(\text{TAP}) = -\frac{\log\left(\frac{\sigma_{ap\lambda1}(\text{TAP})}{\sigma_{ap\lambda2}(\text{TAP})}\right)}{\log(467/652)} \qquad \text{Eq. (16)}$$





$$\text{AAE(CAPS, NEPH)} = -\frac{\log\left(\frac{\sigma_{ap\lambda1}(\sigma_{ep}(\text{CAPS}) - \sigma_{sp}(\text{NEPH}))}{\sigma_{ap\lambda2}(\sigma_{ep}(\text{CAPS}) - \sigma_{sp}(\text{NEPH}))}\right)}{\log(450/630)} \qquad \text{Eq. (17)}$$


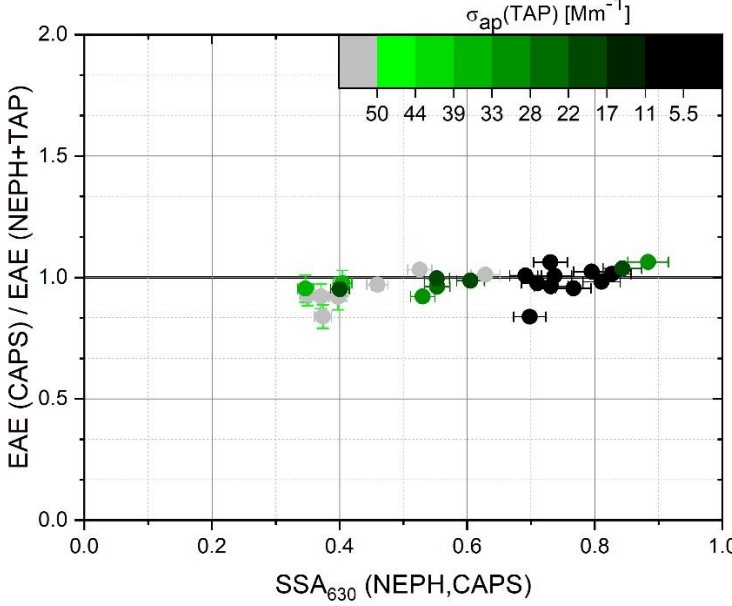

**Figure 9.** This figure shows ratio of the extinction Ångström exponent EAE(CAPS) / EAE(NEPH, TAP) as function of SSA(CAPS, NEPH) at 630nm wavelength and the light absorption coefficient $\sigma_{ap}$(TAP) at 630 nm in the colour code for AQ/AS mixtures.


Neither the SSA, nor $\sigma_{ap}$ show a systematically dependence on the EAE ratios EAE(CAPS) / EAE(NEPH,TAP) . Only method to method ratios for fresh flame soot deviate up to a ratio of 1.5, indicating higher EAE(CAPS) values compared to EAE(NEPH, TAP). Possible explanations for this may be some smaller aerosol bursts which occur while the flame flickers or the particle shape influencing the measurement accuracy. A possible explanation for this could be that variances in a small

timescale are smoothed out by the bigger volume of the Nephelometer compared to the CAPS instrument, where these fluctuations are seen within the timeline.





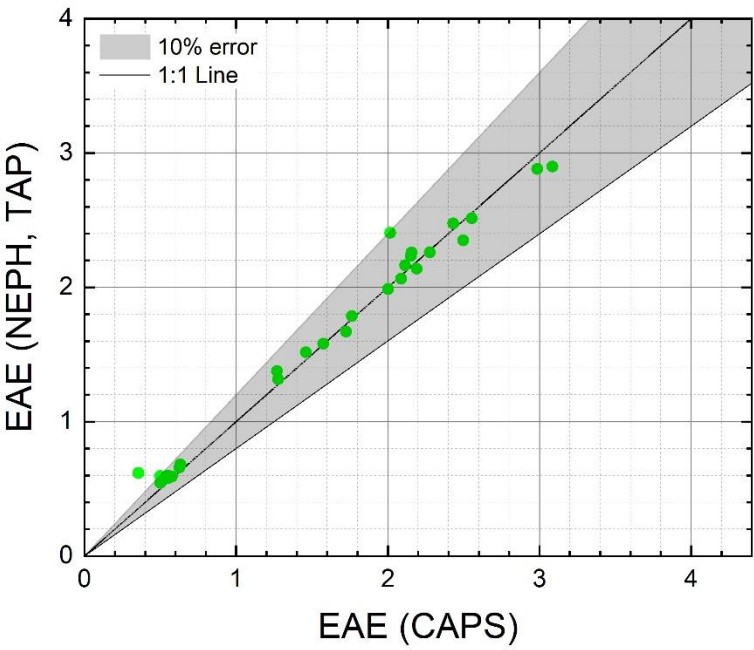

**Figure 10.** Scatter plot of EAE values obtained by CAPS compared to EAE values obtained from TAP + NEPH measurements
showing AQ/AS mixtures.

When directly comparing EAE(TAP, NEPH) to EAE (CAPS) the EAE values agree within 10% deviation. Again, the best
correlation is visible with Aquadag mixture particles. For EAE(CAPS) > 2.5 the EAE(TAP, NEPH) tents to underestimate the
EAE. Nevertheless EAE(NEPH, TAP) shows the highest values close 3.21 which corresponds to EAE values reported in Table
3 for the pure AS particles which are small in size of about 40nm.


2.2.3 Ångström exponents: SAE





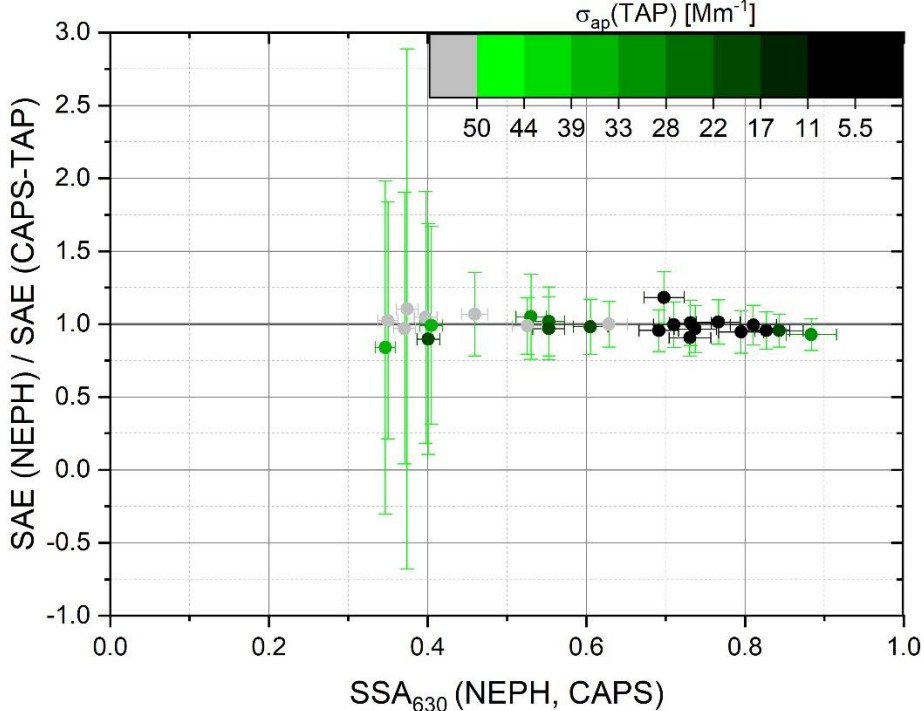

**Figure 11.** This figure shows the instrument-to-instrument ratio of the Scattering Ångström exponent SAE(NEPH) / SAE(CAPS, TAP) as function of SSA(CAPS, NEPH) wavelength and as function of $\sigma_{ap}$(TAP) as a colour code both for at 630 nm.

The instrument-to-instrument ration of SAE were calculated for each particle type. Most data points show a ratio of close to 1 not biased by $\sigma_{ap}$ or by the SSA shown in and Figure 11.

Looking for the instrument-to-instrument SAE ratios for the different absorbing species individually in Table 12 only soot shows an instrument-to-instrument ratio of about 1.43+-0.61 which is statistical not significant different from 1.





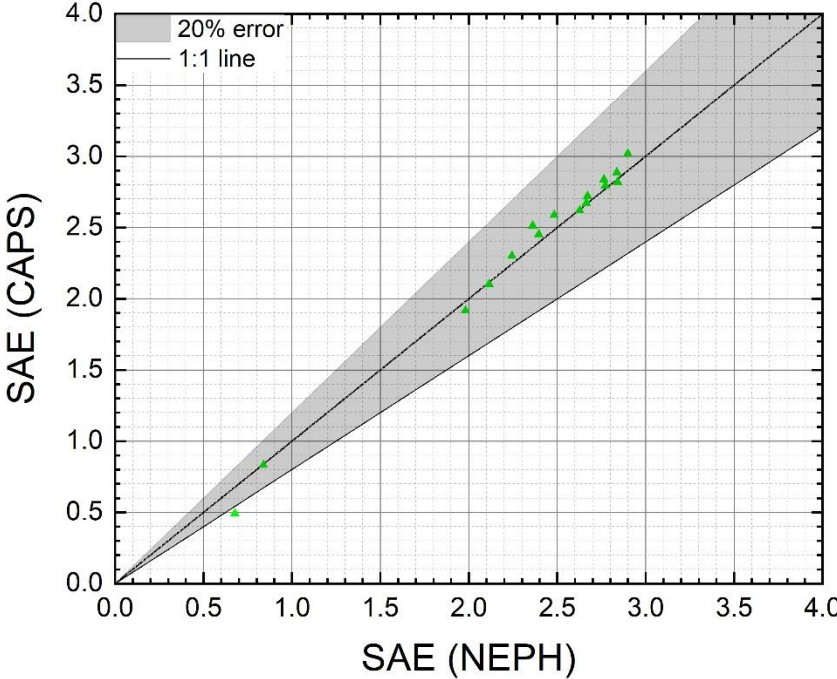

**Figure 12.** Scatter plot of SAE values obtained by CAPS is compared to SAE values obtained by NEPH for AQ/AS mixtures.

When comparing the SAE dataset obtained by using Nephelometer and CAPS is measurements in Figure 12 and Table 12, Aquadag shows the best instrument-to-instrument ratio of 0.99 +-0.15. A small nonlinearity for SAE values higher 3.0 begins to deviate from the 1:1 line but stays within 15% deviation as already seen for EAE. Here again NEPH shows higher SAE values compared to CAPS by a factor 0.9. This factor corresponds as the observed factor for the EAE values and is linked to nephelometer measurements for fine AS particles. Since the Nephelometer correction is calculated based on the scattering

angstrom exponent, which contains a vague size distribution information, it could fail to give correct values for aerosol mixtures and for different sizes.

### 2.2.4 Ångström exponent: AAE

The absorption Ångström exponent depends entirely on the absorbing particle type and should not differ when the light

absorbing particle is mixed with non-light absorbing particles. This independency of adding AS to the mixture was observed for the filter based instrument TAP, visible as x-axes in the scatter plot Figure 13, but when the AAE was calculated using measurements of the in situ instruments Nephelometer and CAPS, the AAE deviates more with increasing SSA and lowering





σ$_{ap}$ shown in Figure 14. For mixtures of AQ the AAE (CAPS, NEPH) is calculated even unphysical negative values. As a result of the error propagation for precision errors, shown as individual error bars in both figures it can be stated that

AAE(CAPS,NEPH) values are not trustworthy especially for AQ mixtures with AAE(TAP) < 1.

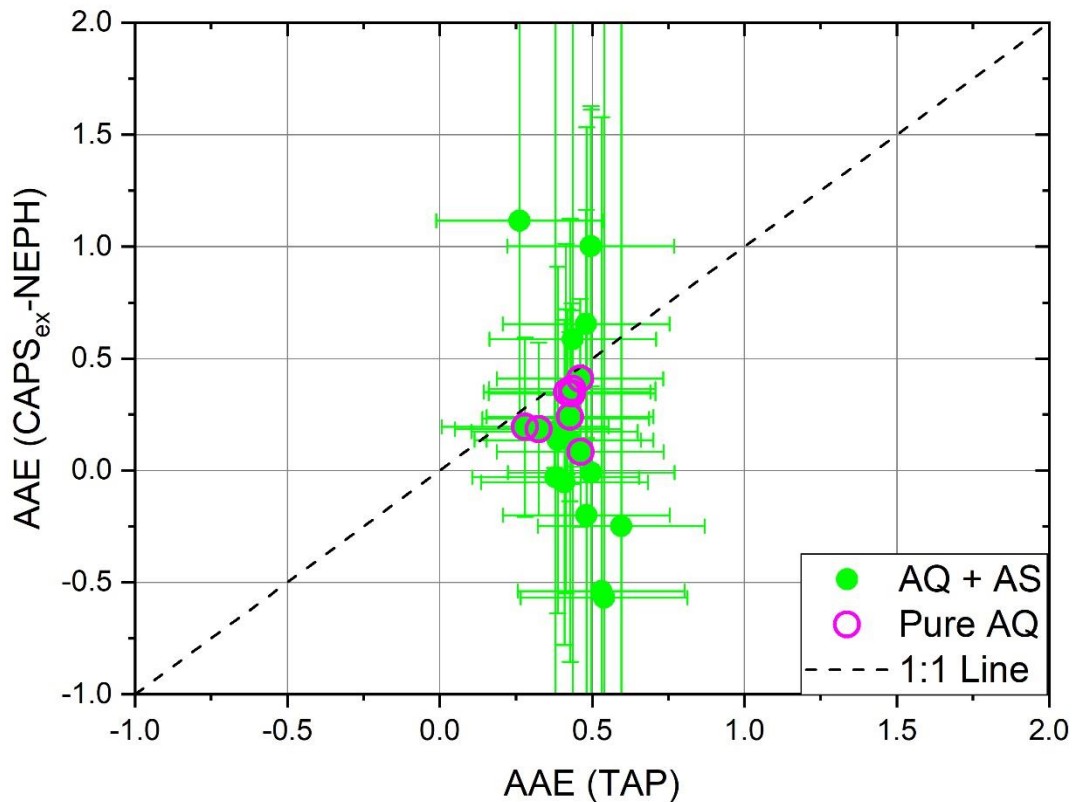

**Figure 13.** Scatter plot of AAE values obtained by AAE(TAP) compared to AAE(CAPS, NEPH).

The reason for this is the high relative precision error associated with low absorption particle loads for AAE (CAPS, NEPH), which we had crosschecked by calculating the variation of the AAE by varying the input variables by their possible max errors, showing the same results. Pure Aquadag particles are made visible by an open circle and it is visible, that those does not stray far from the 1:1 line in Figure 13 including AAE vales for Aquadag. For the pure substance a higher particle load could be used and no other negative interfering non absorbing aerosols influence the measurements.





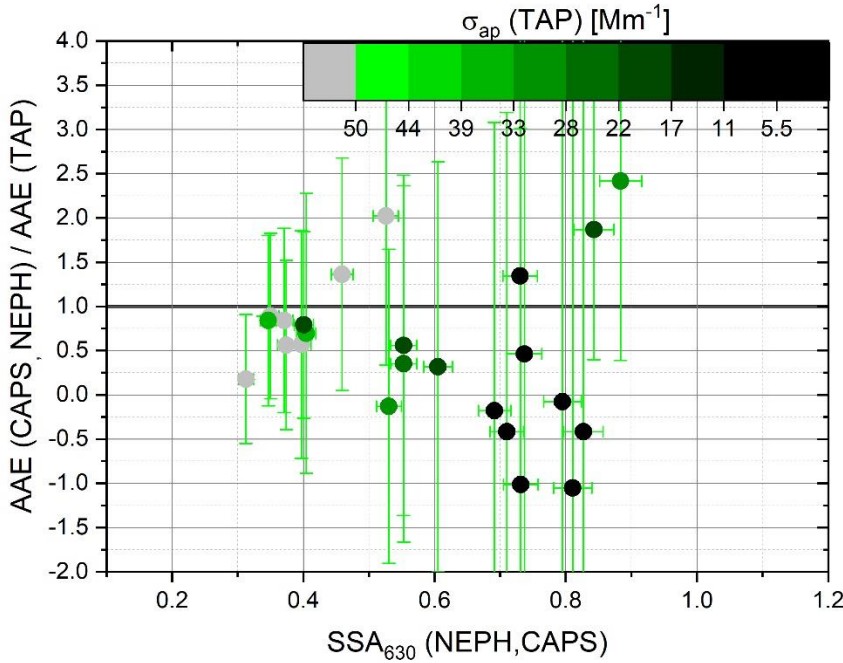


**Figure 14**. This figure shows the instrument-to-instrument ratio of the absorption Ångström exponent AAE(CAPS, NEPH) / EAE(TAP) are shown as function of SSA(CAPS, NEPH) 630nm and information of light absorption coefficient $\sigma_{ap}$(TAP) as a colour code are shown.

In Figure 14 showing the Method-to-method ratios AAE(CAPS, NEPH) / EAE (TAP) there is a strong dependency as function of $\sigma_{ap}$(TAP) and SSA(CAPS,NEPH)_630 visible. Lowering the absorption coefficients below 100 $Mm^{-1}$ or a SSA higher than 0.5, the AAE begins to differ strongly and up to tends up to triple the AAE value calculated from TAP coefficients only. As long as for laboratory studies, these high particle concentrations could be archived, but are rarely present in atmospheric conditions EAE(CAPS, NEPH) method is not applicable for atmospheric measurements


**Table 12.** Ensemble averages for the instrument-to-instrument ratios of the Ångström exponents (EAE, SAE, AAE) using Ångström exponent calculated using instrument calculations and Ångström exponents (EAE, SAE, AAE) using single instrument data as reference.

| Ångström coefficient ratio | BC | AQ | SOOT | MB |
|---|---|---|---|---|
| EAE(Neph,TAP) / EAE(CAPS) | $0.92 \pm 0.07$ | $1.05 \pm 0.15$ | $0.99 \pm 0.56$ | $0.97 \pm 0.15$ |



| SAE(CAPS,TAP) / SAE(Neph) | $1.13 \pm 0.10$ | $0.99 \pm 0.15$ | $1.43 \pm 0.61$ | $1.09 \pm 0.15$ |
|---|---|---|---|---|
| AAE(CAPS,NEPH) / AAE(TAP) | $1.72 \pm 0.85$ | $0.39 \pm 1.70$ | $1.19 \pm 0.93$ | $0.91 \pm 2.32$ |


To compare the overall accuracy of the instrument to instrument rations are compiled for EAE(NEPH,TAP)/ EAE(CAPS), SAE(CAPS,TAP)/SAE(NEPH) and AAE(CAPS,NEPH)/AEA(TAP) are shown in Table 12. Here an ensemble average and the associated variance was considered as a good reference. The instrument to instrument ratios for Ångström exponents for light extinction and Ångström exponents for scattering correspond within 10% deviation. The most prominent exception is

again freshly produced combustion soot. For light absorption, a large deviation for the AAE ratios value is associated with weak absorption coefficients of the mixtures used. Therefore, the AAE shows the biggest differences within the instrument to instrument ratio analysis.

## 3. Conclusion


A major goal of this study was to determine the errors associated with instrumental uncertainties of intensive optical aerosol parameters such as single scattering albedo and Ångström exponents. Basis was an instrument intercomparison study of widely used measurement techniques that are suitable for long-term observations. The methods used agreed the most for a mixture of the spherical-shaped colloidal graphite (Aquadag) as light-absorbing and ammonium sulphate as light-scattering aerosol

component. Results for this mixture have low uncertainties and agree within 10% deviation between the methods for single scattering albedo, extinction Ångström exponent and scattering Ångström exponent. Laj et al.,2020 recently stated requirements for GCOS (Global Climate Observing System) applications. Here he proposed uncertainties lower than the 20% measurement uncertainty for single scattering albedo measurements for attributing and detecting changes to a climate feedback. The uncertainties and deviations shown in this work are with 8-10% measurement uncertainty fulfil the required limit. Overall,

we were able to show study that extensive parameters agree within the limits of uncertainty for the individual instruments. For spherical particles, we achieved the highest correlations for each light extinction, scattering and absorption coefficients. For fractal-like particles, the correlation for light absorption between the in-situ and filter method weakens but stays within instrument uncertainty ranges. Uncertainties increase for intensive parameters, especially for parameters obtained with the differential method that calculates light absorption as the difference between light extinction and light scattering. In addition,

extinction Ångström exponents, scattering Ångström exponents, and single scattering albedo were not as much affected by the uncertainties associated with the differential method used for $\sigma_{ap}$ compared to the absorption Ångström exponent. Using the differential method, AAE was rarely within typical physical values for the differential method. Low single scattering albedo values (<0.5) and, more importantly, high particle loads of at least 50 Mm$^{-1}$ are necessary to reach satisfactory uncertainty levels. Freshly-generated combustion soot differs the most, with results disagreeing up to 30% between filter-based absorption



coefficient data and in-situ methods. This is due to the combined effects of small flickers of the inverted flame generator during the experiment, the overall filter correction schemes, and the physical behaviour of agglomerates. The single scattering albedo for 630 nm wavelength could be determined within 10% deviation between the instrument combinations of CAPS, TAP and Integrating Nephelometer, but tends to differ by at least 0.1 for light absorption coefficients of over 50 Mm$^{-1}$. A similar accuracy could be achieved with a wavelength of 450 nm, for which a 15% deviation between the instrument combinations must be considered. Even with the strong deviation within absorption values, the intensive parameters for the scattering and extinction Ångström exponent stay within 10% deviation, regardless which instrument combination is used for calculation. With this approach the intensive aerosol properties showed a high rate of agreement between different instrument sets for the determination of these properties for techniques used for long-term measurements, except for the absorption angstrom exponent. As an additional result, we can present that for stable aerosol production, the internal scattering coefficient measurement by the CAPS PM$_{SSA}$ agrees with the integrating Nephelometer within 10% deviation and therefore could be substitute the TSI Nephelometer 3563 for light scattering measurements which is not produced any longer.

*Acknowledgments*. Parts of this work were supported by IAGOS-D (Grant Agreement No. 01LK1301A), EU H2020 Project ENVRIplus (Grant No.654182) and HITEC Graduate School for Energy and Climate. A special thanks to Paola Formenti for her assistance with the manuscript.

*Contributions of co-authors*. PW performed all instrument calibrations, the instrumental set up, and the data analysis. UB and BF designed the LabVIEW environment of the experimental set up. MB helped during instrument preparations. AF and TO provided technical details of the instrumentation. PW, OB, UB and AP contributed to the manuscript and the interpretation of the results.

*Conflict of interest*. The authors declare that they have no conflict of interest.

Data availability.

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
