# Peer review of "Relative errors of derived multi-wavelength intensive aerosol optical properties using CAPS PMSSA, Nephelometer, and TAP measurements"

_Atmospheric Measurement Techniques, 2021_

## Author Comment (AC1)

The paper presents results of laboratory measurements of aerosol optical properties measured with a CAPS PMssa, a nephelometer and a filter-based absorption photometer, the TAP. Different types of BC particles were produced with a nebulizer and a burner. Also, purely scattering aerosols (ammonium sulfate) were nebulized. The data were used for calculating scattering and absorption, single-scattering albedo and Ångström exponents of extinction, scattering and absorption. The results show that scattering, absorption and extinction and their wavelength dependencies - except AAE - can be measured in the aerosol phase with the CAPS PMssa with a reasonably low uncertainty. This is very valuable because all filter-based instruments have artifacts in the absorption measurements. It is also valuable that different types of BC particles and size distributions were used in the experiments.

The paper is well written and I can recommend publishing it in AMT after some additions and answers to some questions that puzzle me.

Comments and questions

1) Compared to ambient measurements the concentrations were fairly high. The best results for the absorption coefficient with the CAPS PMssa were observed for absorption coefficients > 10 Mm-1 . Such levels are observed in very polluted environments, for instance in China and India. I think you could mention this and also refer to a new intercomparison where the CAPS PMssa was used in real background conditions, see Asmi et al. (2021) (ref below) and also compare your results with theirs.

**Answer:**

**We will reference Asmi et al in the paper. The Pallas station is at clean artic air conditions, which results in low aerosol light absorption levels. That paper shows, that DM methods (EMS called there) are at the detection limit even after one hour of averaging. This result is in accordance with our results. We see some issues with Asmi at al. concerning inlet setup and truncation correction but in principle we can state that the CAPS Instrument performs well for Ext. Sca and SSA at rural conditions like previously stated by Onasch et al 2015, Massoli et al. 2010. Four our own measurements at low absorption coefficient conditions for AQ (5 Mm-1) we have calculated an error of the DM method of up to 2 Mm-1 (compared to TAP 0.2 Mm-1).**
**If no direct absorption measurement is available, the use of CAPS SSA gives you also a fair measurement of the absorption coefficient. The situation is different for the AAE calculation with a reported value of AAE=0.4 +- 1.7. Thus, our conclusion for rural sites is that AAE calculations based on DM method using CAPS must be taken with caution. We clarified this in the text.**

2) Related to this, I am missing a table where you would show the extensive and intensive aerosol optical properties and the length of each experiment. Maybe in a supplement? Now the tables have various ratios and regression constants – which is important of course – but I think it would be useful to show also the range of absorption and scattering coefficients you have produced. Or if you don't want to make that supplement, at least you could add some lines to the overview table, Table 3: number of experiments, average length of experiments, averages and ranges of scattering and absorption coefficients.

**Answer:**

**Table 3 shows the measured and calculated values for the pure substances. We add the information about the ranges for absorption/ extinction of the different experiments in the experimental section. Here the extinction coefficients are ranging from 15-2000 Mm$^{-1}$ (all aerosol Types) (0-1000 Mm$^{-1}$ for light absorption coefficient of mixed aerosol types). We have aimed for**

**extinction values ranging from 30 to 100 Mm$^{-1}$ to stay at atmospheric relevant values. We will add information about the ranges of scattering and absorption in the tables 7,8 and 9.**

3) The results in the scatter plots and the tables are based on experiment averages or ensemble averages and their ratios. As an example I take Table 5. I do not find anywhere information of how long data are collected for one experiment's average. In the table there are the numbers N=xx. I suppose xx the number of experiments, right? Please explain clearly both in the text and the table captions what N means.

**Answer:**

**We will clarify on Page 13 in the Extensive Parameter Section that for each experiment (run) a different aerosol mixture was generated with different overall extinction levels. We used a 100 sec average (1 HZ sampling rate) per experiment (run), after the nephelometer reached a steady state. In Figures 3-5 100 second averages and standard deviations are plotted. Table 5 compiles EMS (DM) derived absorption coefficients. Here we report ensemble averages, where N denotes the number of experiments (run) used for the average.**

4) Further on the same averaging question. So, the scatter plots are based on averages which I assume means using the average of each experiment. How would the results change, if you used shorter averaging times, from some seconds to minutes? Or was the aerosol production so stable that it would not matter, which averaging time was used? Usually it is assumed that the uncertainty due to noise decreases with one over the square root of averaging time. This then propagates to the uncertainties of the derived optical properties. For instance, I guess that in Fig. 6 the data points would fill in the grey shaded error bands if shorter averaging times were used. Discuss this a bit.

**Answer:**

**We agree with the reviewer on this important point. We chose to average 100 seconds of the 1 HZ data per experiment based on reported Allan standard deviation results. The Allen Plots in Massoli et al., 2010 for CAPS extinction measurements (http://dx.doi.org/10.1080/02786821003716599) show that there is a minimum detection limit close to 100 sec. For longer integration times, the measured variance starts to increase again due to baseline drift. Baseline drift is corrected in the CAPS by taking zeros at a user-defined frequency. In this study, we chose to measure zero baselines for all CAPS every 10-12 minutes, thus a 100 second average makes sense. A similar result is observed for the TAP/CLAP instrument shown in Ogren et al., 2017 Figure 7. In this study, Ogren observed the TAP/CLAP variance reached a minimum level of 10% for the aerosol loadings (comparable to our study) after 100 seconds. For rural, ambient measurements, Ogren et al recommend a longer averaging time than 100 seconds.**

Lines 68 – 73. There is discussion on AAE. It is written that AAE depends on chemical composition and that AAE > 1 is due to brown carbon or mineral dust. This is not the whole truth. It is easy to show with your Mie code and it has been shown in several papers that AAE also depends on the size

distribution of the light absorbing particles and that both AAE>1 and AAE<1 values are observed even for pure BC particles. Here are just some references: Gyawali et al. (2009), Lack and Cappa (2010), Lack and Langridge (2013), Liu et al. (2018), Zhang et al. (2020) and Virkkula (20219. Actually, it is interesting that if you compare median diameters the AAE values in your Table 3 with Fig. 6 of Liu et al. (2018) they seem to be qualitatively in agreement

**Answer:**

**We agree with the reviewer's comment. Our intent was to state that for the conditions we used in this study (i.e., solid absorbing particles externally mixed with solid non-absorbing particles), the AAE is expected to be near unity.  We will make this clear in the text in on page 3 in the introduction section and will add your suggestions.**

Line 112: " Because all instruments were connected to one central aerosol supply line." The sentence should continue, now there is a full stop.

**Fixed. Thank you.**

Line 136: "Data inversion for the nephelometer …". Is "inversion" really the correct term here? The scattering coefficients are simply multiplied with a correction factor that depends on SAE.

**We changed "inversion" to "data correction".**

Line 199: Size distributions were measured beforehand. Why not all the time? Any idea of the stability of the size distributions?

**The size distributions of the aerosol production single particle types (i.e., absorbing, or non-absorbing) were characterized before the study and checked several times during the different experiments. Size distributions just relate to the MFC setup. The Size distribution is not expected to change during one experiment if the solution centration and flows do not change.  (Liu, Aerosol generator of high stability, 1975)**

References
 Great, thank you for your efforts!

Asmi et al.: Absorption instruments inter-comparison campaign at the Arctic Pallas station, Atmos. Meas. Tech., 14, 5397–5413, 2021.

Gyawali et al.: In situ aerosol optics in Reno, NV, USA during and after the summer 2008 California wildfires and the influence of absorbing and nonabsorbing organic coatings on spectral light absorption, Atmos. Chem. Phys., 9, 8007–8015, 2009.

Lack and Cappa: Impact of brown and clear carbon on light absorption enhancement, single scatter albedo and absorption wavelength dependence of black carbon, Atmos.Chem. Phys., 10, 4207–4220, 2010.

Lack and Langridge: On the attribution of black and brown carbon light absorption using the Ångström exponent, Atmos. Chem. Phys., 13, 10535–10543, 2013.

Liu et al: The absorption Ångström exponent of black carbon: from numerical aspects, Atmos. Chem. Phys., 18, 6259–6273, 2018.

Zhang et al:  The absorption Ångstrom exponent of black carbon with brown coatings: effects of aerosol microphysics and parameterization, Atmos. Chem. Phys., 20, 9701–9711, 2020.

Virkkula: Modeled source apportionment of black carbon particles coated with a light-scattering shell,  Atmos. Meas. Tech., 14, 3707–3719, 2021

---

## Author Comment (AC2)

This paper compares uses from different combinations of instruments to assess uncertainties in spectral optical properties (both measured (scattering/absorption/extinction) and derived (single scattering albedo and scat/abs/ext Angstrom exponents). They find that using the differential method (i.e., ext-scat) for absorption is very unprecise/uncertain at absorption loading levels <20 Mm-1 (the wavelength of absorption for that constraint is not stated). The uncertainty leads to uncertain absorption Angstrom exponents (AAEs). The authors recommend only using the differential method to obtain AAE when absorption > 50 Mm-1. This suggests that the differential method is only applicable in the most polluted atmospheric conditions for absorption-related parameters (see for example Figure 5 in Laj et al., (2020) which suggests that even GAW monitoring sites in urban areas rarely reach absorption loading of 20 Mm-1 at 550 nm). In contrast, the greater robustness of extinction and scattering observations means the wavelength dependence of these properties (e.g., scattering and extinction Angstrom exponents) are trustworthy at extinction levels ~20 Mm-1 (though again the wavelength for this constraint is not stated)? The authors suggest that either method they use for calculating single scattering albedo (CAPS PMssa or nephelometer+filter-based absorption instrument) is within the desired uncertainty for SSA.

General comments

While the authors are primarily focused on assessing the differential method for intensive aerosol optical properties in a lab setting, it would greatly expand the relevance of their paper if they could comment more on the implications of their findings for the many long-term monitoring sites around the globe that make aerosol optical property measurements (e.g., ACTRIS network). I'm not 100% sure that's a fair question as, in their study, the authors haven't considered actual atmospheric aerosol in all of its infinite variety (e.g., other absorbing species such as dust and 'brown carbon'; internal mixtures; etc.), but those limitations could be acknowledged in such a discussion. Filter-based measurements of absorption are often dismissed as lacking (e.g., Lack et al., 2008) https://www.tandfonline.com/doi/full/10.1080/02786820802389277), but here it seems they perform quite well.

1) **Answer:**

**The paper (Lack et al., 2008) shows a small aerosol loading range and compares the PSAP with a PAS method. For a monotype aerosol it shows well agreeing data between 0-35 Mm$^{-1}$ for light absorption. When comparing those with highly polluted environmental data, those differ by nearly 60% and a "R" of 0.8. We will imply those variety and limitations you mentioned. We will clarify that our study was done for „simple "externally mixed solid absorbing and solid non-absorbing particles over the extinction range from 15 to 150 Mm-1. Thus, this study does not explicitly address real-world ambient aerosols that can be internally or externally mixed or both, contain particles with liquid, solid, and semi-solid phases, and may contain multiple sources of absorbing material. Thus, the results of this study cannot directly comment on any of the issues you mentioned. That said, this study does find closure for these commonly used instruments for the particle types and mixture studied like Lack et al did find for monotype aerosol in the lab.**

Please have the manuscript read/corrected by native English speaker or technical writer with eye for detail.

**2) Answer:**

**Two of our coauthors are native speakers, they will take care on this.**

Science comments

The Abstract suggests that EAE and SAE are trustworthy for extinction coefficient values > 20 Mm-1. That result is not supported by anything presented in the main manuscript - none of the figures show SAE coloured by extinction and extinction coefficient loading is not mentioned in the discussion of SAE. (Figure 11 shows the SAE coloured by absorption not extinction.) Please elaborate. What are the results that lead to this conclusion? Extinction at what wavelength? and for what averaging time?

**3) Answer:**

**The Absorption Value was given as a colour code at 630 nm. We wanted to show, that adding light absorbing aerosol loading does not influence the instruments response for the SAE. We would agree that the extinction dependency is missing here We have not included these plots explicitly in the current text but will add them in the annex or supplement.**

Line 115 - presumably then the time resolution of the other instruments was also reduced to 10 min? or was only 10min data recorded for the nephelometer while the highest resolution for the other instruments was used and then averaged to 10min when comparisons involved the nephelometer?

**4) Answer:**

**All instruments recorded data at a 1 second rate. We kept the aerosol in the line in a steady state, which had a fluctuation measured with the OPC, CAPS PMssa and NEPH below 2% deviation. We decided to average these 1Hz data for all instruments for intercomparison within the 100sec after the nephelometer becomes stable. We clarify this in the text.**

Line 165++ Error propagation - I refer you to the supplemental materials in Sherman et al., 2015 (www.atmos-chem-phys.net/15/12487/2015/) where the error calculations for these variables (xAE and SSA) and for scattering and absorption are described in detail.

**5) Answer:**

**Thanks for the recommendation. As suggested by the BIPM (Bureau International des Poids et Mesures) our calculations do not include error propagation of the wavelength correction and**

calibration assumptions because these effects are of minor importance and as long the calibration is done properly. But we will cite the proposed refences for completeness.

Line 211 - the uncertainty of absorption measured by the CLAP depends on the aerosol SSA and to a lesser extent on the averaging time. See equation 7 and figure 9 in Ogren et al. (2017). The Mueller paper does not consider the CLAP (or TAP). Are the PSAP uncertainties/precisions reported in Mueller assumed to be the same?

**6) Answer:**

**It was assumed, that filter based measurements share issues due filter artefacts, SSA and backscattering. As visible in the Ogren 2017 paper, the uncertainty for the averaging times and scattering, mostly effects values below 1-5 Mm-1 and stays afterwards at 10% uncertainty. The Correction algorithm is mentioned for the PSAP in the Müller Paper based on the Ogren 1998 Paper and then evolved with Virkulla 2010 which is the same for the CLAP /TAP. CLAP and PSAP are the comparable instruments from the measurements principle and for used filter materials. In addition, the same correction functions are applied. In an Intercomparison campaign (Asmi et al. 2021) the PSAP reaches the best detection limits, but it is still in the same order compared to all other filter-based methods.**

Related to the two notes above - for what averaging times are the uncertainties in the manuscript calculated? Would your results/conclusions change if longer averaging times were considered? I ask because long-term monitoring sites (e.g., GAW sites) typically use nephelometers and filter-based absorption photometers to obtain extensive and intensive optical properties. The GAW sites typically report hourly averaged data.

**7) Answer:**

**Longer averaging periods strengthens data sets with absorbing values close the detection limit, like Asmi et al 2021 demonstrated. Short time events like plumes by traffic planes, etc. are not visible in the time series after averaging. The Allen plots by Massoli (2010) and Ogren (2017) show that after 100 seconds of averaging the lowest variance and thus the lowest detection limit is reached. Averaging for longer periods would decrease TAP, CAPS variance since the long-term drifts due to transmission (TAP) and baseline drift (CAPS) causes increased variance values.**

Line 206 'Thus, the SAE drops to 0,76 for 130 nm AQ particles.' I think the SAE drop is more due to the larger contribution of particles due to the spread in the size distribution for AQ than to the median diameter. Collaud Coen et al. (2007) (https://agupubs.onlinelibrary.wiley.com/doi/epdf/10.1029/2006JD007995) show no spectral dependence for 130 nm particles in their Figure 7. AQ (and soot) size distributions range almost to 1000 nm in figure 2. Why does the SAE not drop as much for soot particles which have a slightly larger median diameter than the AQ?

**8) Answer:**

**The shape of AQ is assumed to be more compact than the soot agglomerates, such that their scattering and electrical mobility behaviors are dependent mainly upon their physical diameter. In contrast, the scattering behavior of the fractal soot agglomerates is due mainly to the distribution of primary particles, whereas their electrical mobility diameter is more dependent upon the major axis of the agglomerate. I think you are right with the influence of the larger AQ that drops the SAE, while the "optical" diameter of the soot may be a different than their measured mobility diameter. We will include this discussion in the text.**

Line 243 and Line 489 - make clear that the particles are externally mixed in the experiments (if that is the case). I think the AAE would vary if the absorbing particle was internally mixed with purely scattering aerosol.

9) **Answer:**

**The Particles are external mixtures. Two separate nebulizers were used, as well as two separate dryer tubes. After that they are mixed in a chamber. Thus, the AAE for all mixtures should depend only on the absorbing aerosol distributions and be independent of the non-absorbing particles. We agree with the reviewer; if the non-absorbing material coated the absorbing particles, the AAE would likely be dependent upon both the absorbing and non-absorbing distributions. We have made this clearer in the corresponding text**

Editing/wordsmithing comments

I started making editorial comments but quickly decided that too many were needed and that I should focus on the science. This is an important paper: please have a technical editor and native English speaker spend some time on it to make it as clear and accessible as possible.

**Thank you for your eye to detail and to your desire to help improve our manuscript.**

Line 44 'trough' to 'through'

**That slipped through**

Line 48 'TAP' to 'CLAP' Ogren et al. (2017) was only about the CLAP. The TAP is based on the design of the CLAP but there are some differences related to spot size, flow rate and potentially filter used. Please make this more clear.

**Acknowledged. We have made this clearer in the text by stating the similarities of PSAP and CLAP/TAP design.**

Line 50 add Ogren, 2010, Bond et al., 1999. Also, Virkkula et al., 2010 is the corrigendum to Virkkula et al., 2005 which actually describes the full methodology used to develop corrections to the PSAP. Please cite both Virkkula papers.

**We now include both references.**

Line 63-65 It is unclear what references go with what use of Angstrom exponent. please reorganise sentence. Use semi-colons between different clauses and group Foster and Angstrom references within the same parentheses if they are both for wavelength adjustment.

**We have rewritten this section to make it clearer.**

Line 75 'classify' to 'classifying'

**Changed.**

Line 90 - should Ammonium in Ammonium sulphate be capitalised?

**No, ammonium sulphate should not be capitalized.**

Line 90 'where' to 'were'

**Changed.**

Line 90-91 Rephrase these two sentences - I think really what is meant is that the ammonium sulphate liquid solution concentrations were not changed so that the dry aerosol size distributions would remain constant.

**We have rephrased these two sentences to read, "We kept the ammonium sulphate liquid solution concentrations constant thus, the dry aerosol size distributions would remain constant."**

*Line 99 'Downstream the production the aerosol' to 'Downstream of production, the aerosol' or 'Downstream the production aerosol'*

**We changed the line to read, "Downstream of the aerosol production, the aerosol was injected in a mixing chamber assuring homogenous mixing"**

Line 101 'connected to using' to 'connected using'

**Changed.**

Line 103 - Strange line over period.

**Removed.**

Line 106 - Both 'was used' and 'we used' are used in the manuscript. Please be consistent with passive or active voice but not both.

**We have gone back through the manuscript and changed it all to the active voice.**

line 106-107 'small-sized' suggests there are other sizes of TAPs. Recommend removing 'small-sized' - there are no comments on sizes of other instruments.

**Removed.**

Line 107 - give manufacturer of PSAP

**We now include the manufacturer of the PSAP.**

Line 120 - give refractive index (indices) of PSLs?

**We have added information on the refractive index (1.59) in the text**

Line 123 - 'validating the same factor' - what factor?

**Here we were referring to the geometric factor of the CAPS sampling cells. We have made this clear by stating: [..] calibrated with $CO_2$ for further validation of the same geometric factor for the CAPS sampling cells.**

Line 123 - the neph was directly calibrated with CO2 (and filtered air), correct? The way the sentence is written it sounds like a calibration 'factor' for CO2 was derived from the CAPS and applied to the neph. Please clarify.

**We have rewritten this sentence to read: The NEPH was calibrated using CO2 (Anderson and Ogren, 1998; Modini et al., 2021).**

Line 126-127 - says truncation is not necessary for particles < 200nm but Figure 2 shows significant presence of particles > 200 nm for some species. Make clear that that comment applies to the CAPS. For the neph the Anderson and Ogren truncation will range from 13% (450 nm, SAE=0.76) to 2% (700nm, SAE=3). Onasch et al. (2015) suggests the truncation characteristics of the CAPS PMssa are similar to those of commercial nephelometers, so perhaps CAPS truncation correction is necessary?

**The nephelometer truncation was done by using the schemes by Anderson and Ogren or the Massoli et al (2009) schemes for mixtures and light absorbing aerosols, which ranged from 1 to 6 % over our study. The particle load majority is consistence of ammonium sulphate, which is smaller than 200 nm thus there is no truncation correction needed Onasch et al (2015)**

Line 126 - 'since highest' to 'since the highest'

**Changed.**

Line 126 - 'amount were' to 'amount was'

**Changed.**

Line 127-128 - 'drifting shift of baseline' to 'drifting baseline'   Also - presumably no data were used during the warm up period? Please clarify.

**Changed.  We have clarified in the text on page 5 that only stable data was used in the averages.**

Line 127 - 'Onasch et al., 2015a' to 'Onasch et al., 2015'

**Changed.**

Line 131 - filter spot does not need to be capitalised, remove comma after 'selected'

**Changed.**

Table 1 - should cite both the original Virkkula paper in 2005 and the 2010 Virkkula paper, which is the correction to the original paper.

**We now include both references.**

Line 136 - I would not call the Anderson and Ogren correction a data inversion - the scattering coefficient is multiplied by a correction value C where C=a+b*SAE where a and b are constants and SAE is the uncorrected scattering Angstrom exponent for that particular scattering data point.

**We have changed this to read, "Data correction..."**

Line 136 - nephelometer does not need to be capitalised

**Changed.**

line 136 - 'alterned' to 'altered'

**Changed.**

Line 139 - what is PMex?  presumably you mean sigma_ep?

**Changed to read, "Extinction data from the CAPS PMssa instrument were used without further correction, except for the adjustment factor determined by $CO_2$ measurements and PSL to MIE calculations."**

Line 141 - same comment as above: should cite both the original Virkkula paper in 2005 and the 2010 paper which is the correction to the original paper.

**Done.**

Line 141 - what filter was used with the TAP?  The Virkkula 2005;2010 and Bond 1999 and Ogren 2010 corrections were all done with the Pallflex filter. I believe the TAP is shipped with Azumi filters from Brechtel.  See discussion in Ogren et al 2017 about differences in correction for the different filters.  (Perhaps that is taken care of by the TAP software, but should still be stated what filters were used.)

**The software has the capability to choose the correct correction scheme based on the filter type used.  We have made this clear in the text by describing the software feature and by adding the information on the Filter (BT-TAP-FIL100, ENVILYSE).**

line 143 - 'Probertites' to 'Properties'

**Changed.**

Equation 3 and line 158 - use the same variable as in equation 3a - i.e., sigma_xp.  You could also put an x in front of AE:  xAE= -log(...)

**Changed.**

Line 178 - 'with minimised size distribution chances' - unclear what is meant - do you mean 'with minimal size distribution changes'?

**The CMD and GSD of the ammonium sulphate nebulized by the constant output atomizer depends on the concentration of the salt solution (and the volumetric flows) (Figure 11 of Liu et al 1975; • DOI:10.1080/0002889758507357). We have changed the text to read, "In order to vary the aerosol concentration with minimized size distribution changes, the mixture was controlled by a MFC-controlling the extractive flow after the dehydration tube. The CMD and GSD of the ammonium sulphate nebulized by the constant output atomizer depends on the concentration of the salt solution and flows the atomizer is operated with.  By keeping the solution and the flows constant also the resulting size distribution will remain constant (Liu et al. 1975).**

Line 187 - 'its spherically shape' to 'it is spherically shaped'.  Split this sentence into two or maybe three sentences. It is hard to follow.

**We have changed the text to read, "The results of the intercomparison of Aquadag is expected to be best described by Mie theory, since it has a spherical shape. Therefore, the correction schemes applied to the instruments apply best, since calibration is done by PSL spheres (polystyrene latex beads).  PSL solutions were treated the same as all other aerosol solution samples and their size was confirmed by DMA and OPC measurements."**

Line 189 - 'approved by DMA and OPC.' to 'confirmed by DMA and OPC measurements.'

**Changed.**

Table 3 - Change table title to 'Overview of the aerosol types used and measured parameters'

**Changed.**

Line 205, 206 - Inconsistent notation for numbers - sometimes European notation, but elsewhere use American notation.  Should be consistent.

**We have changed all number notations to American notation (i.e., the decimal is designated with a period).**

Line 212 - extra parenthesis, also ACP standard for refs is (Anderson and Ogren, 1998; Massoli et al., 2009).

**Changed.**

Figure 2 caption - change to: 'Size distributions measured by DMA and CPC ...), Also, use a lighter blue for soot - hard to distinguish from the black.

**We have changed the caption and we now use a lighter blue for the soot.**

Line 228 - 'data points averages of at least 100 seconds' - earlier it's noted that the neph time resolution was 10 min (6000 seconds).  Please clarify.

**As previously stated,: All instruments recorded data at a 1 second rate. We kept the aerosol in the line in a steady state, which had a fluctuation measured with the OPC, CAPS PMssa and NEPH below 2% deviation. We decided to average these 1Hz data for all instruments for intercomparison within the 100sec after the nephelometer becomes stable. We clarify this in the text.**

Line 233 - 'for 450nm ... wavelength' to 'for the 450nm ... wavelengths'

**Changed.**

Figure 3 - are there points with SSA > 0.9? I recommend using a different colour for those points than the error band colour. Could a consistent SSA gradiant be used? - the SSA steps vary between 0.07 and 0.08 in the colour code legend. (Also the colour scale for SSA appears to change amongst the plots - e.g., for figure 3 the highest SSA value is 0.90, while for figure 4&5 it's 0.90 and 0.95. The figure 3 caption suggests that the colour code represents the SSA at 630 nm even if the plot is for 450 nm. Please correct if that is not the case.

**Thanks for the advice. We have changed the colour so its distinguishable. Originally, we actually displayed the 630 nm SSA even for 450 nm datapoints using the colour code, because the SSA for 630 nm is much broader. We have now changed the 450 nm data color code to represent the actual SSA values obtained at 450 nm.**

Line 262 - 'nor a strong shift for high or low volumetric cross-section values' do you mean loading?

**Indeed. We have changed the text to read, "There is neither a dependence of the mixture ratio with ammonium sulphate (which the SSA is the indicator) visible, nor a a dependency as function of light absorbing aerosol loadings"**

Line 294 - tables should be numbered in the order in which they are mentioned in text. Here Tables 7-9 are referred to before Tables 5 and 6.

**Sorry, the reference where not updated by word for unknown reasons. Thanks for finding this mistake**

Line 296 - 'delivers reliable SSA' - the SSA results have not been reported yet in this paper. Should this statement have a citation?

**We moved this statement to the SSA section (Intensive Parameters) where we compare them to the CAPSssa to CAPS/NEPH SSA results.**

Line 300 - '1 Hz' measurement - but the neph is averaged to 10 min which is not 1 Hz.

**The Instruments deliver measurement points at 1 Hz. The Nephelometer time resolution is indeed way higher, but as soon the steady state is reached, the measurement points could be treated as such.**

Line 310 - is the variance for sigma_ap<10 Mm-1 not shown? Table 5 only shows ensemble and variance > 10 Mm-1? for what wavelength is the filtering done?

**The statement was that when values below 10 Mm-1 are ignored, the ensemble average stabilizes as well as its deviation. The Variance observed for absorption coefficients less than 10 Mm-1 are much higher. The filtering was done for 630 nm wavelength.**

For both Table 5 (630 nm) &Table 6 (450 nm) is the sigma_ap>10 Mm-1 variance for the wavelength that the table is for (i.e., 630 nm for Table 5 and 450 nm for Table 6)? or do both tables assume variance for wavelength > 630 nm?

**If one of the absorption coefficients regarding 450 or 630nm wavelength exceeds 10 Mm-1 both absorption coefficients were excluded from the experiment. that's why the number of experiments N is the same for both wavelengths. After that the variance was calculated for the wavelength mentioned in the title of the tables.**

Line 316 - 'As a result, this increase also the errors associated with the differential method' Is this saying that the higher loading is leading to higher uncertainty in absorption?

**Scattering and absorption measured at 450 nm is typically higher than 630 nm. Thus, for the same aerosol mixture, the 450 nm data should have higher signal levels. We saw higher deviations for lower wavelength and the DM/EMS. We will clarify this accordingly.**

Line 320 - for what wavelength is the filtering done?

**We have changed the text to read, "Again filtering the 450 nm data for ..."**

Line 385 - why is the CAPS(scat)/NEPH(ext) combo used as the SSA reference rather than CAPS(scat)/CAPS(ext) which would measure in the same volume and not have the flow/averaging issues of the neph? Onasch et al. (2015) lists the advantages that the SSA from the CAPS PMssa has over calculating SSA from two separate instruments.

**Our intention was to test the CAPS against an independent reference. Thus, it makes no sense to take the CAPS_SSA as reference. We clarified this in the text.**

Line 385 - 'often used combination' - often used by who?

**The DM/EMS Method using cavity ringdown extinction coefficients measurements and nephelometer scattering coefficients measurements as refence (x-axis) is used for example in P.Sheridan et. Al 2005.**

Figure 7 - as suggested above - use different colour for points than is used for the error band.

**Good suggestion. We have changed the colors for points over 50 Mm-1 enhancing the visibility**

Figure 9 and Figure 10 are not referred to in the text.

**We have changed the text : Neither the SSA, nor $\sigma_{ap}$ show a systematically dependence on the EAE ratios EAE(CAPS) / EAE(NEPH,TAP) visible in Figure 9. […] When directly comparing EAE(TAP, NEPH) to EAE (CAPS), the EAE values agree within 10% deviation, visible in Figure 10.**

Figure 10&12 - why are points not coloured by loading?

**We have added a color scale to indicate loading. Even with this information, the Points align with the 1:1 Line suggestion that there is no loading influence.**

Line 482-483 'Here again NEPH shows higher SAE values compared to CAPS by a factor 0.9.'  Please check. Both Figure 12 and Table 12 suggest the CAPS SAE is typically higher than the NEPH SAE - most CAPS points are above the 1:1 line and the CAPS/neph ratios in Table 12 are greater than 1.  Also, where does the factor of 0.9 come from?

**The factor /difference of 10% was earlier shown at the light scattering comparison for aquadag, which is just there for aquadag and not for the other aerosol types.**

Figure 14 - 'EAE(TAP)' to 'AAE(TAP)'.   Are these the same points as are shown in Figure 13?

**Yes, these are the same points.  We changed the text to read "AAE (TAP)**

Line 514 - 'EAE(TAP)' to 'AAE(TAP)'.

**Changed.**

Line 547-549 - make clear that the constraints 'Low single scattering albedo values (<0.5) and, more importantly, high particle loads of at least 50 Mm-1' are specific to the differential method and don't appear to be relevant for the filter-based method of obtaining AAE.

**We have changed the text to read, "Low single scattering albedo values (<0.5) and, more importantly, high particle loads of at least 50 Mm$^{-1}$ are necessary to reach satisfactory uncertainty levels for the DM"**

Line 551 - the conclusion states that the largest disagreement for absorption coefficient is due in part to filter correction schemes, but this was not shown/discussed in the manuscript.

**We state the largest disagreement for the flame soot.  which is in most complex aerosol type regarding morphology we used. For more compact particles, the scattering is stronger (Radney et al 2014). A stronger backscattering is not considered in the correction schemes which might be responsible for the disagreement. We will clarify this in the text.**

Line 561 - The suggestion to use the CAPS PMssa as a substitute for the TSI nephelometer doesn't seem logical to me - first - three CAPS would be needed to provide the same spectral coverage at 3 wavelengths that the TSI nephelometer provides in one instrument.  Second, the CAPS PMssa doesn't provide backscattering coefficient.  The Ecotech Aurora 3000 (or 4000) nephelometer seems to be a more reasonable TSI nephelometer replacement as it provides the same functionality as the TSI instrument.  If someone already has a CAPS then yes - it could be used as a stand-in for the TSI neph total scattering for one wavelength, but I would hesitate recommending a CAPS in lieu of a nephelometer - it would depend on the scientific question that was being addressed.

**We have changed the text to read, "As an additional result, we can present that in our study, the scattering coefficient measurement by the CAPS PM$_{SSA}$ agrees with the integrating Nephelometer within 10% deviation. Therefore, it could be substituting the TSI Nephelometer 3563 for light scattering measurements since it is not produced any longer.**

This does not mean that nephelometer are not of interest any more because the provide additional information on backscattering information at multiple wavelength in one instrument.